

# Broad similarities in shoulder muscle architecture and organization across two amniotes: implications for reconstructing non-mammalian synapsids

Philip Fahn-Lai[1], Andrew A. Biewener[2] and Stephanie E. Pierce[3]

[1] Museum of Comparative Zoology, Concord Field Station and Department of Organismic and Evolutionary Biology, Harvard University, Cambridge, MA, USA
[2] Concord Field Station and Department of Organismic and Evolutionary Biology, Harvard University, Cambridge, MA, USA
[3] Museum of Comparative Zoology and Department of Organismic and Evolutionary Biology, Harvard University, Cambridge, MA, USA

Corresponding authors
Philip Fahn-Lai,
phillai@g.harvard.edu
Stephanie E. Pierce,
spierce@oeb.harvard.edu

## ABSTRACT

The evolution of upright limb posture in mammals may have enabled modifications of the forelimb for diverse locomotor ecologies. A rich fossil record of non-mammalian synapsids holds the key to unraveling the transition from "sprawling" to "erect" limb function in the precursors to mammals, but a detailed understanding of muscle functional anatomy is a necessary prerequisite to reconstructing postural evolution in fossils. Here we characterize the gross morphology and internal architecture of muscles crossing the shoulder joint in two morphologically-conservative extant amniotes that form a phylogenetic and morpho-functional bracket for non-mammalian synapsids: the Argentine black and white tegu *Salvator merianae* and the Virginia opossum *Didelphis virginiana*. By combining traditional physical dissection of cadavers with nondestructive three-dimensional digital dissection, we find striking similarities in muscle organization and architectural parameters. Despite the wide phylogenetic gap between our study species, distal muscle attachments are notably similar, while differences in proximal muscle attachments are driven by modifications to the skeletal anatomy of the pectoral girdle that are well-documented in transitional synapsid fossils. Further, correlates for force production, physiological cross-sectional area (PCSA), muscle gearing (pennation), and working range (fascicle length) are statistically indistinguishable for an unexpected number of muscles. Functional tradeoffs between force production and working range reveal muscle specializations that may facilitate increased girdle mobility, weight support, and active stabilization of the shoulder in the opossum—a possible signal of postural transformation. Together, these results create a foundation for reconstructing the musculoskeletal anatomy of the non-mammalian synapsid pectoral girdle with greater confidence, as we demonstrate by inferring shoulder muscle PCSAs in the fossil non-mammalian cynodont *Massetognathus pascuali*.

## BACKGROUND

The differences separating therian locomotion from that of other extant quadrupeds are usually understood as a contrast between derived "erect" or "parasagittal" vs. plesiomorphic "sprawling" limb posture (*Bakker, 1971*) (Fig. 1). Mammal-like posture is associated with adducted limbs, joints aligned in a single plane, dorsoventral bending of the axial skeleton, and the ability to use asymmetrical gaits (*English, 1978*; *Biewener, 1991*; *Fischer, 1994*; *Fischer & Blickhan, 2006*; *Carrier, Deban & Fischbein, 2006*; *Bonnan et al., 2016*). On the other hand, sprawling posture features abducted limbs, multiaxial joints, mediolateral axial bending, and mostly symmetrical gaits (*Sukhanov, 1974*; *Edwards, 1977*; *Peterson, 1984*; *Ritter, 1992*; *Ashley-Ross, 1994*; *Reilly & Elias, 1998*; *Blob & Biewener, 1999*; *Baier & Gatesy, 2013*). While kinematic studies of "erect" mammals (*Jenkins, 1971b*) and "sprawling" non-mammals (*Nyakatura et al., 2019*) have shown enough variation within each locomotor archetype to cast doubt on the usefulness of a rigid postural framework (*Gatesy, 1991*), it remains the case that mammalian limb use is uniquely diverse among amniotes. Mammals inhabit a wide range of habitats, using modified limbs to run, climb, swing, swim and fly. The forelimbs, in particular, have been transformed almost beyond recognition in many cases, not only becoming wings and flippers, or elongated with reduced digits and hooves, but also finding use in various non-locomotor behaviors such as prehension, excavation, grooming, and manipulation (*Polly, 2007*; *Vaughan, Ryan & Czaplewski, 2013*).

If exaptation of the forelimb was an important factor in the adaptive radiation of mammals, the groundwork for mammal-like posture and locomotion must have been laid down earlier, along the synapsid stem. However, despite a rich fossil record, major gaps remain in our understanding of non-mammalian synapsid musculoskeletal function. Our ability to interpret function is limited by the fact that the joints of the pectoral skeleton became unconstrained early on in synapsid evolution, shortly after the Permian emergence of the therapsid clade (*Kemp, 2005*). Differing hypotheses (*Jenkins, 1970*; *Bakker, 1975*; *Kemp, 2005*; *Lai, Biewener & Pierce, 2018*) concerning non-mammalian synapsid postural evolution have, hence, yet to find unequivocal support based on skeletal evidence alone, highlighting the need to consider the role of unfossilized soft tissues such as muscles. But before the muscle function of extinct vertebrates can be analyzed, it must first be reconstructed. To date, this has largely been accomplished using the extant phylogenetic bracket (*Witmer, 1995*). By correlating fossilized skeletal morphology with observed hard and soft tissue states in extant outgroup animals, this technique has been used to parsimoniously infer muscle organization in fossils ranging from basal sarcopterygian fish and stem tetrapods (*Romer, 1922*; *Miner, 1925*; *Molnar et al., 2018a*, *2018b*) to early reptiles (*Holmes, 1977*), bird-line archosaurs (*Gatesy, 1990*; *Dilkes, 1999*; *Hutchinson & Gatesy, 2000*; *Carrano & Hutchinson, 2002*; *Jasinoski, Russell & Currie, 2006*; *Langer, Franca & Gabriel, 2007*; *Holliday, 2009*; *Maidment & Barrett, 2011*; *Burch, 2014*; *Persons, Currie & Norell, 2014*), and indeed, mammalian as well as non-mammalian synapsids (*Watson, 1917*; *Gregory & Camp, 1918*; *Jenkins, 1971a*; *Cox, 1972*; *Kemp, 1980a*, *1980b*; *Walter, 1986*;

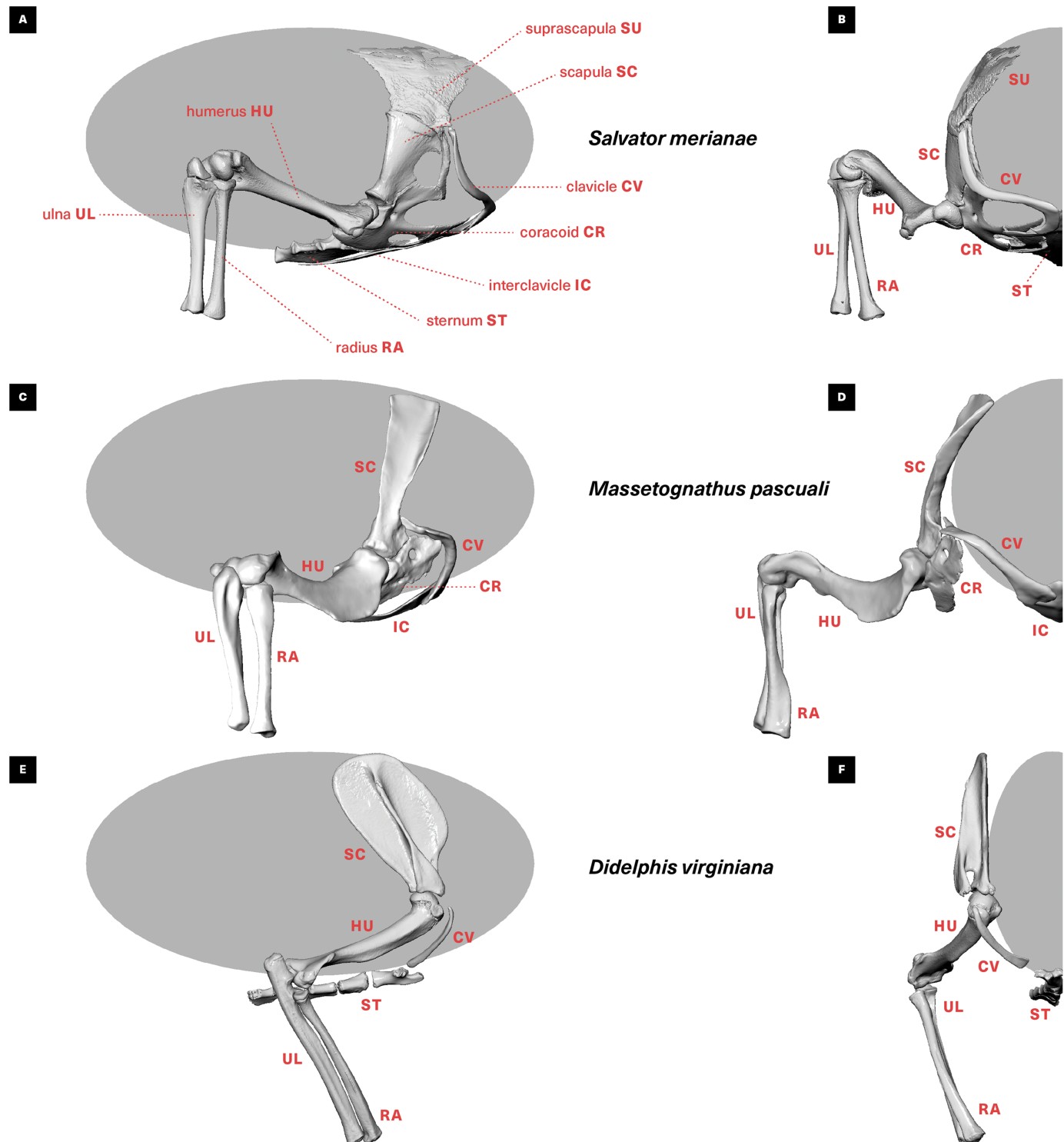

**Figure 1 Skeletal anatomy and midstance posture of (A and B) "sprawling" *Salvator merianae*, (C and D) *Massetognathus pascuali*, and (E and F) "erect" *Didelphis virginiana* pectoral limb.** All animals shown in right lateral view (A, C, E) and cranial view (B, D, F). *M. pascuali* is depicted in a hypothetical neutral pose, reflecting the middle of each joint's range of motion (see *Lai, Biewener & Pierce, 2018*). *S. merianae* and *D. virginiana* poses taken from 3D videoradiography. While in *Massetognathus* the ventral portion of the scapulocoracoid comprises separate procoracoid and metacoracoid elements (sensu *Vickaryous & Hall, 2006*), here they are collectively termed the "coracoid" for simplicity.

*Ray & Chinsamy, 2003*; *Oliveira & Schultz, 2016*; *Cuff, Goswami & Hutchinson, 2017*; *Lai, Biewener & Pierce, 2018*).

Extant phylogenetic bracketing sheds light on the presence or absence of muscles in fossil taxa, as well as on the locations of their attachments. While knowledge of each muscle's origin and insertion sites provides insight into its general action and mechanical advantage (*Maidment et al., 2014*; *Otero et al., 2017*), a muscle's functional characteristics are determined not only by its attachments, but also by the internal architectural organization of its fibers in relation to its tendon. The geometry of a muscle's contractile and elastic elements strongly influence its mechanical output and working range by altering the component of force and length change directed along its line of action (*Gans, 1982*; *Lieber, 2002*). Hence, two equal-sized muscles with identical attachments but differing architectural properties might have very different functional characteristics, complicating the interpretation of muscles reconstructed on the basis of skeletal anatomy alone (*Bates & Schachner, 2012*; *Bates & Falkingham, 2018*). Finding a way to reconstruct muscle architecture would greatly strengthen reconstructions of posture and locomotion in fossil synapsids, as it would provide a means to estimate input parameters for computational musculoskeletal models and test competing postural/locomotor hypotheses (*Hutchinson, 2004*; *Nagano et al., 2005*; *Bates et al., 2012*; *Charles et al., 2016*; *Regnault & Pierce, 2018*; *Bates & Falkingham, 2018*).

Unlike attachments, muscle architecture cannot be rigorously inferred using the extant phylogenetic bracket, since internal fiber organization has no known skeletal correlates. Instead, extant animals that bracket the morphological and functional extremes of the fossil group of interest may help constrain reconstructions of architecture. The evolutionary history of non-mammalian synapsids saw not only a transition from "sprawling" to "erect" limb posture, but also a dramatic transformation in body proportions. The early "pelycosaurian"-grade synapsids had robust girdles, undifferentiated axial skeletons and relatively short limbs, but by the Jurassic, the first mammaliaforms had reduced pectoral girdles and regionalized axial skeletons, as well as proportionately longer limbs (*Kemp, 2005*). These morphological contrasts reflect the differences in gross proportions between extant lizards and therian mammals. Quantifying and comparing muscle architecture between representatives of these extant amniote groups should therefore furnish a baseline picture of architectural variation across two contrasting morphological and functional paradigms, and help bookend reconstructions of architecture in analogous or intermediate non-mammalian synapsids.

In this study, we present an in-depth qualitative and quantitative comparison of shoulder musculature in two amniotes that comprise an extant phylogenetic bracket as well as morpho-functional analogues for extinct non-mammalian synapsids: the Argentine black and white tegu *Salvator merianae* and the Virginia opossum *Didelphis virginiana* (Fig. 1). These two living amniotes are comparable in size, life history, and unspecialized ecology, yet differ greatly in the use of their limbs. As such, they represent suitable archetypes for investigating the functional morphology of "sprawling" and "erect" posture, respectively. We focus on muscles crossing the shoulder, a mechanically-important joint that is considered central to distinguishing between mammalian and non-mammalian

posture (*Romer, 1922*; *Gray, 1944*). By combining nondestructive diffusible iodine-based contrast-enhanced computed tomography (diceCT) (*Gignac et al., 2016*) with traditional physical dissection in an experimentally-controlled context, we provide a comprehensive picture of shoulder musculoskeletal anatomy in two morphologically-conservative, comparably-sized amniotes. Our aim is to establish a framework for estimating architectural muscle parameters for extinct stem synapsids that will strengthen future biomechanical musculoskeletal models of extant and fossil taxa, ultimately yielding fundamental new insight into the acquisition of mammal-like posture and locomotion.

## MATERIALS AND METHODS

### Two study species

We collected anatomical data from six wild adult Argentine black and white tegus (*S. merianae*) and six wild adult Virginia opossums (*D. virginiana*) (Tables S1 and S2). While varanid and iguanian lizards have traditionally been used as models of plesiomorphic amniote posture and locomotion (*Jenkins & Goslow, 1983*; *Padian & Olsen, 1984*; *Ritter, 1996*; *Blob & Biewener, 1999*; *Blob, 2000*; *Farlow & Pianka, 2000*; *Clemente et al., 2011*; *Dick & Clemente, 2016*), tegus were chosen here to represent a more shallowly-nested clade of terrestrial generalists (*Sheffield et al., 2011*; *Simões et al., 2018*) of growing importance as laboratory animals (*Bennett & John-Alder, 1984*; *Montero et al., 2004*; *Toledo et al., 2008*; *Sheffield et al., 2011*). In comparison, among extant mammals, didelphid opossums are a well-established plesiomorphic model for therian development, anatomy, and locomotion (*Broom, 1899*; *Jenkins, 1971b*; *Hiiemae & Crompton, 1985*; *Klima, 1985*; *Parchman, Reilly & Biknevicius, 2003*; *Sánchez-Villagra & Maier, 2003*; *Gosnell et al., 2011*; *Hübler et al., 2013*; *Diogo et al., 2016*; *Bhullar et al., 2019*).

Similarities in life history facilitate direct comparison between *Salvator* and *Didelphis*. Both animals are opportunistic omnivores of similar size, with high growth rates and high fecundity. Both are active foragers capable of sustained locomotion, with basal metabolic rates that overlap during the warmer spring and summer months (*Urban, 1965*; *Fournier & Weber, 1994*; *Toledo et al., 2008*). Finally, individuals of both species are easily available; *S. merianae* is invasive to southeastern Florida (*Pernas et al., 2012*; *Jarnevich et al., 2018*), while *D. virginiana* is commonplace throughout North and Central America (*McManus, 1974*). These traits make tegus and opossums ideal animal models for exploring anatomical and functional adaptations for "sprawling" and "erect" limb posture in amniotes (*Butcher et al., 2011*; *Sheffield et al., 2011*; *Gosnell et al., 2011*).

### Muscle identification, topology and architecture

Here we focused on muscles that spanned the glenohumeral joint of the shoulder. Extrinsic muscles such as m. trapezius, m. serratus, and mm. rhomboidei were not considered, nor were muscles with only soft tissue attachments such as m. dorsoepitrochlearis. Preliminary homology hypotheses were taken from the literature (*Diogo et al., 2009*), then expanded based on our findings from digital and physical dissections.

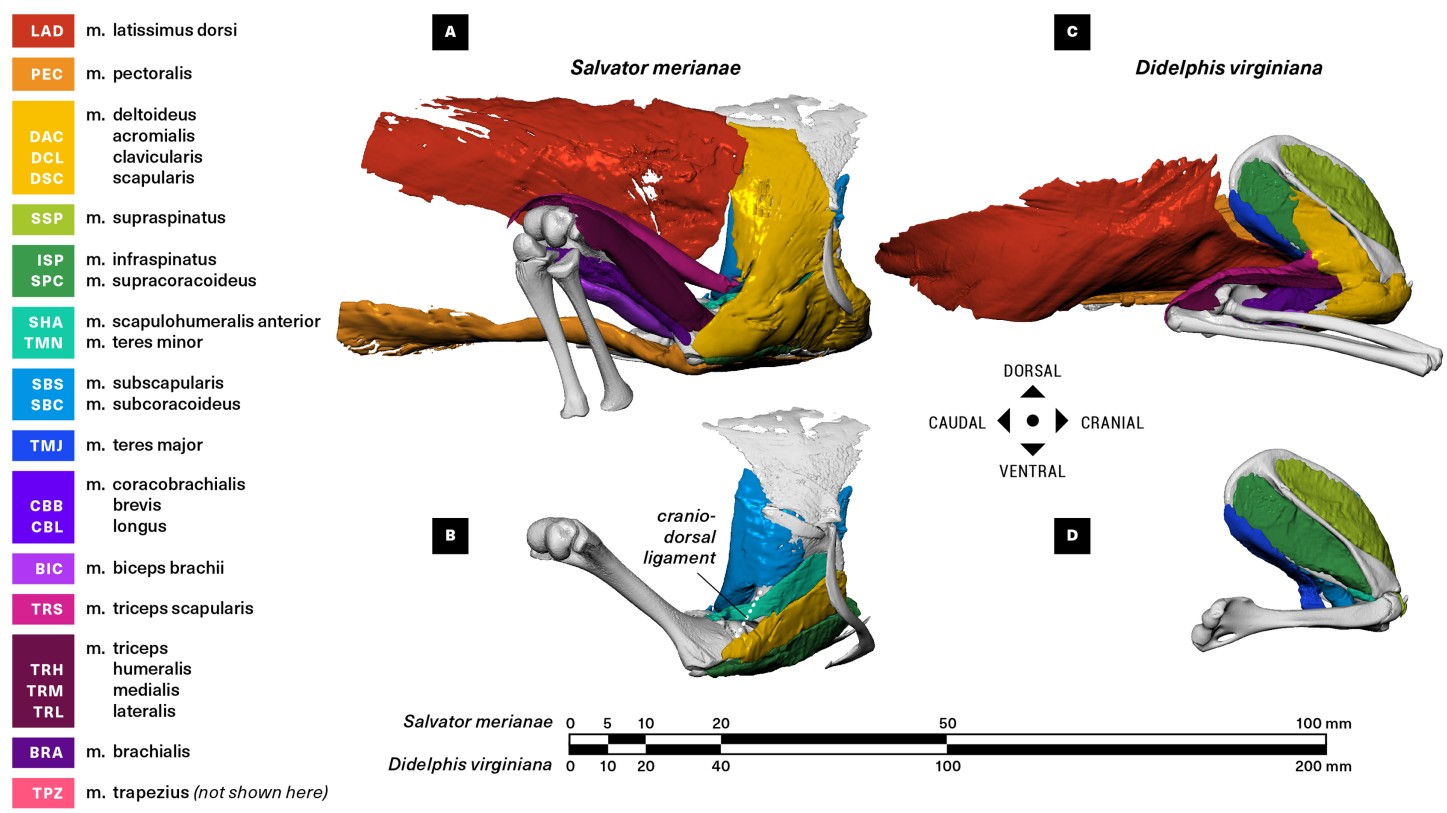

| | | |
|---|---|---|
| **LAD** | m. latissimus dorsi | |
| **PEC** | m. pectoralis | |
| **DAC** **DCL** **DSC** | m. deltoideus acromialis clavicularis scapularis | |
| **SSP** | m. supraspinatus | |
| **ISP** **SPC** | m. infraspinatus m. supracoracoideus | |
| **SHA** **TMN** | m. scapulohumeralis anterior m. teres minor | |
| **SBS** **SBC** | m. subscapularis m. subcoracoideus | |
| **TMJ** | m. teres major | |
| **CBB** **CBL** | m. coracobrachialis brevis longus | |
| **BIC** | m. biceps brachii | |
| **TRS** | m. triceps scapularis | |
| **TRH** **TRM** **TRL** | m. triceps humeralis medialis lateralis | |
| **BRA** | m. brachialis | |
| **TPZ** | m. trapezius *(not shown here)* | |

**Figure 2** **Right lateral view of pectoral girdle and proximal forelimb of the Argentine black and white tegu (*Salvator merianae*) (A and B) and the Virginia opossum (*Didelphis virginiana*) (C and D).** Showing all muscles crossing the shoulder joint (A and C), and the deep layer of muscles originating on the scapulocoracoid/scapula (B and D).

## Digital dissection: skeletal morphology and in situ muscle topology

Two cadaveric tegus (Body Mass 0.68 and 1.04 kg) and two opossums (Body Mass 1.04 and 1.11 kg) were skinned, gutted and fixed for 24 h in 10% neutral-buffered formalin solution. Following fixation, each animal was first subjected to a baseline X-ray microcomputed tomography (μCT) scan to capture skeletal morphology. Per published diceCT guidelines, the animals were then rinsed with deionized water and completely immersed in Lugol's iodine solution (I₃K) to increase the radiodensity of muscle and other soft tissues, rendering them visible to μCT (*Gignac et al., 2016*). As these were relatively large animals with few precedents in the diceCT literature, we experimented with specimen treatment, immersion times and stain concentration throughout protocol development.

One tegu and one opossum were beheaded and bisected midway between the last rib and the pelvis, while the other two individuals were stained as whole animals. Each animal was posed with one forelimb extended and maximally protracted and the other flexed and maximally retracted, in order to bracket the limits of ex vivo mobility (Fig. 2). One tegu was stained in a 10% Lugol's solution for one week, while the other tegu and the two opossums were stained in 3% Lugol's for three, six, and nine weeks respectively. Once a week, each specimen was removed from the iodine solution, rinsed in deionized water, and μCT-scanned to check progress. Once satisfactory contrast enhancement was

achieved, each animal was imaged a final time using a Nikon Metrology (X-Tek) HMXST225 scanner (Nikon Metrology, Tokyo, Japan) at the Harvard University Center for Nanoscale Systems. μCT technique was specimen-specific for optimal image quality, but fell within the following parameter ranges: 115–130 kV, 61–130 μA, 1,000–2,000 ms exposure, tungsten target, 0.5–1 mm Cu filter, 127 $\mu m^3$ voxel size.

The raw μCT data were reconstructed in Nikon CT Pro 3D as DICOM series, and imported into Mimics v19 (Materialise NV, Leuven, Belgium) for segmentation. Muscles and bones were identified and manually isolated with two-dimensional masks, and three-dimensional surface meshes were then computed and exported as .stl files. We used the landmark-based point registration tool in 3-Matic v11 (Materialise NV, Leuven, Belgium) to align clean bone meshes from the baseline scans with the noisier post-stain bone meshes. We used the mesh editing tools in MeshLab (ISTI-CNR, Pisa, Italy) and Autodesk MeshMixer (Autodesk Inc., San Rafael, CA, USA) to downsample and retriangulate all meshes. The refined meshes were then imported into Autodesk MudBox (Autodesk Inc., San Rafael, CA, USA), and muscle attachments (origin/insertion) were painted directly onto bone models using the PTEX mapping method (Figs. 3–6). The locations and shapes of muscle attachment areas were later manually verified in separate physical dissections.

## Physical dissection: muscle architectural properties

To validate muscle attachment sites and to measure muscle architectural properties, we skinned and dissected four additional cadaveric tegus (Body Mass 1.33 ± 0.11 kg) and four opossums (Body Mass 1.35 ± 0.26 kg). Muscles were identified and photographed in situ to validate the attachment areas previously determined from diceCT scans, then carefully excised, photographed again, dipped in 1x phosphate-buffered saline, blotted dry, and weighed. Each muscle-tendon unit (MTU) was weighed (g) with and without external tendon. Each muscle was then incised along its line of action and photographed again. ImageJ (U.S. National Institutes of Health, Bethesda, MD, USA) was used to measure fascicle length (mm) and internal pennation angle ($\theta$, °). Measurements were repeated three times each at three separate locations in the muscle. To address measurement error, we collected measurements bilaterally whenever possible (i.e., unless one of the sides was too damaged) and took the average. Finally, we calculated each muscle's physiological cross-sectional area (PCSA)—a proxy for maximum isometric force—using a standard formula (Eq. (1); *Sacks & Roy, 1982*), substituting the commonly-used literature value of 0.001056 $gmm^{-3}$ for the density of typical vertebrate muscle (*Mendez & Keys, 1960*). As Eq. (1) generalizes to both pennate and non-pennate muscles (*Lieber & Fridén, 2000*), we followed common practice in applying it uniformly across all of our muscles (*Allen et al., 2010*; *Dick & Clemente, 2016*; *Cuff et al., 2016*).

$$PCSA = \frac{\text{Muscle mass} * \cos\theta}{\text{Fascicle length} * \text{density}} \qquad (1)$$

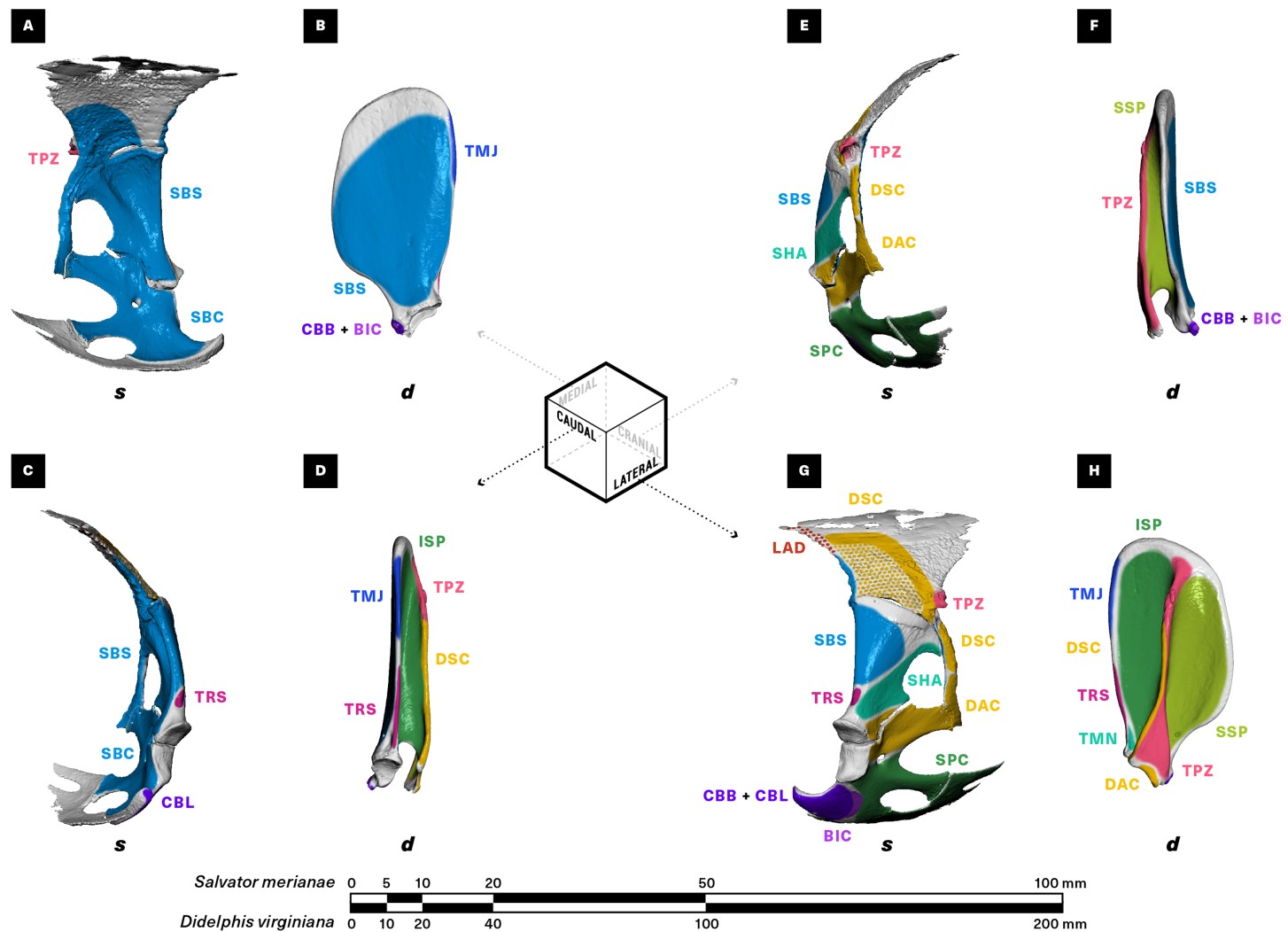

**Figure 3 Muscle attachments on the right scapulocoracoid/scapula of *Salvator merianae* (s) and *Didelphis virginiana* (d).** Shown in medial (A and B), caudal (C and D), cranial (E and F), and lateral (G and H) view. Stippled areas represent loose fascial associations between muscle and bone. Muscle abbreviations and color-coding follow Fig. 2.

To facilitate comparisons between the two species, we assumed isometry within the small size range of the specimens studied and normalized each individual's measurements to a 1 kg animal, dividing lengths by body mass$^{1/3}$, areas by body mass$^{2/3}$, and masses by body mass. Species means and standard deviations were computed for muscle mass ($M_m$, g), MTU mass ($M_{mtu}$, g), muscle length ($L_m$, mm), MTU length ($L_{mtu}$, mm), fascicle length ($L_f$, mm), pennation angle ($\theta$, °), and PCSA (mm$^2$) (Table 1). Differences in normalized mean $M_m$, PCSA, $L_f$, and $\theta$ between homologous muscles were investigated using unpaired two-sample Student's *t*-tests (all variables were normally distributed based on Shapiro–Wilk testing at $\alpha = 0.05$), and the results were corrected for multiple simultaneous statistical comparisons (Fig. 7; Table S3). Finally, normalized PCSA was plotted against normalized $L_f$ to uncover gross trends in muscle function across the evolutionary bracket.

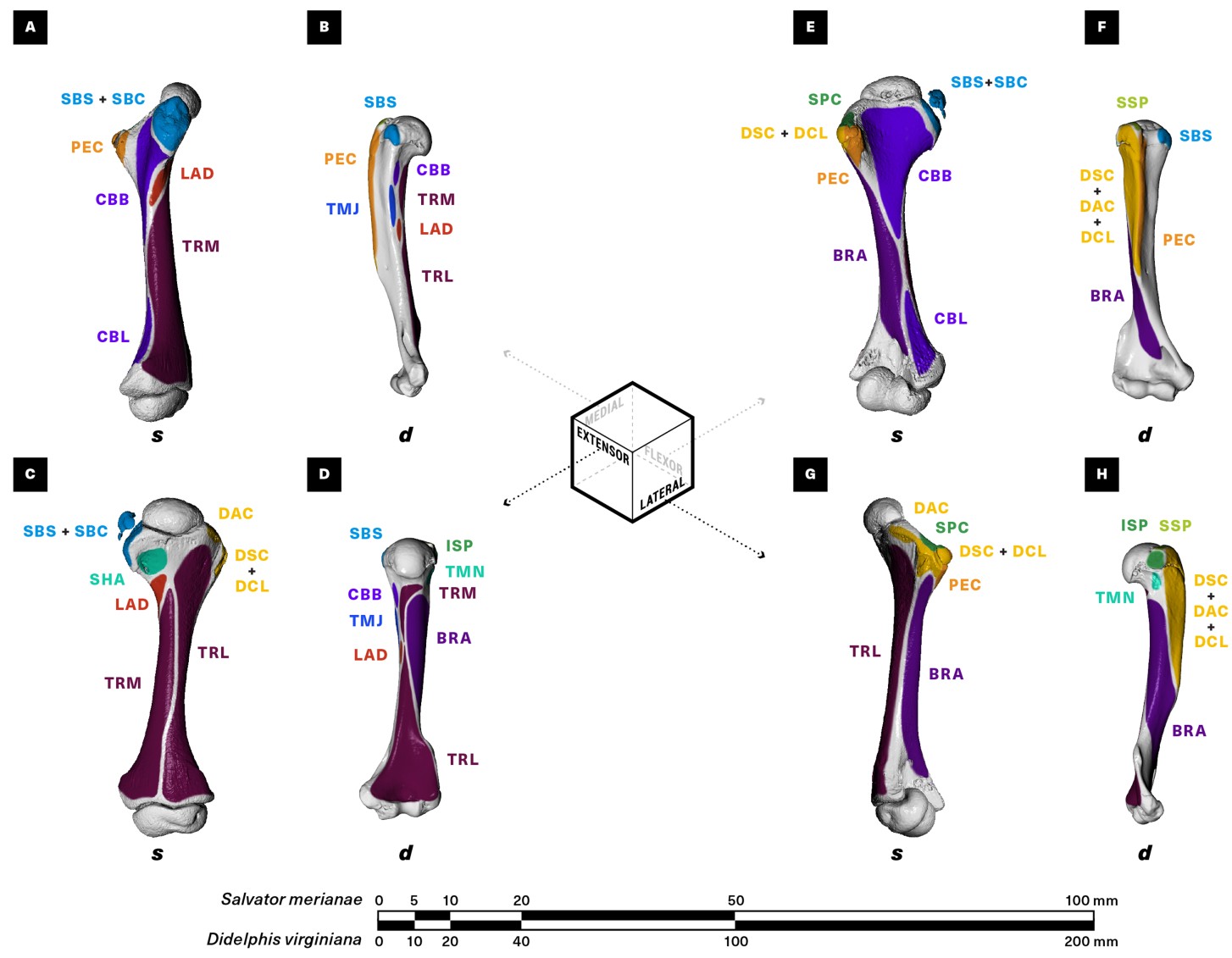

**Figure 4 Muscle attachments on the right humerus of *Salvator merianae* (s) and *Didelphis virginiana* (d).** Shown in medial (A and B), extensor (C and D), flexor (E and F), and lateral (G and H) view. Muscle abbreviations and color-coding follow Fig. 2.

## RESULTS

In order to more easily compare morphology between a "sprawling" and an "erect" quadruped, we use a muscle-centric convention for orienting limb bones. The dorsally-oriented surface of the tegu humerus and the more caudally-oriented surface of the opossum humerus are primarily covered by the triceps complex, which acts to extend the elbow; we refer to these as the "extensor" surfaces of their respective humeri. Similarly, the ventrally-oriented surface of the tegu humerus and the cranio-ventrally oriented surface of the opossum humerus are covered by the elbow flexors m. brachialis and m. biceps brachii and are referred to as "flexor" surfaces. The remaining surfaces are referred to as "medial" and "lateral" based on their distance from the body wall. This convention also

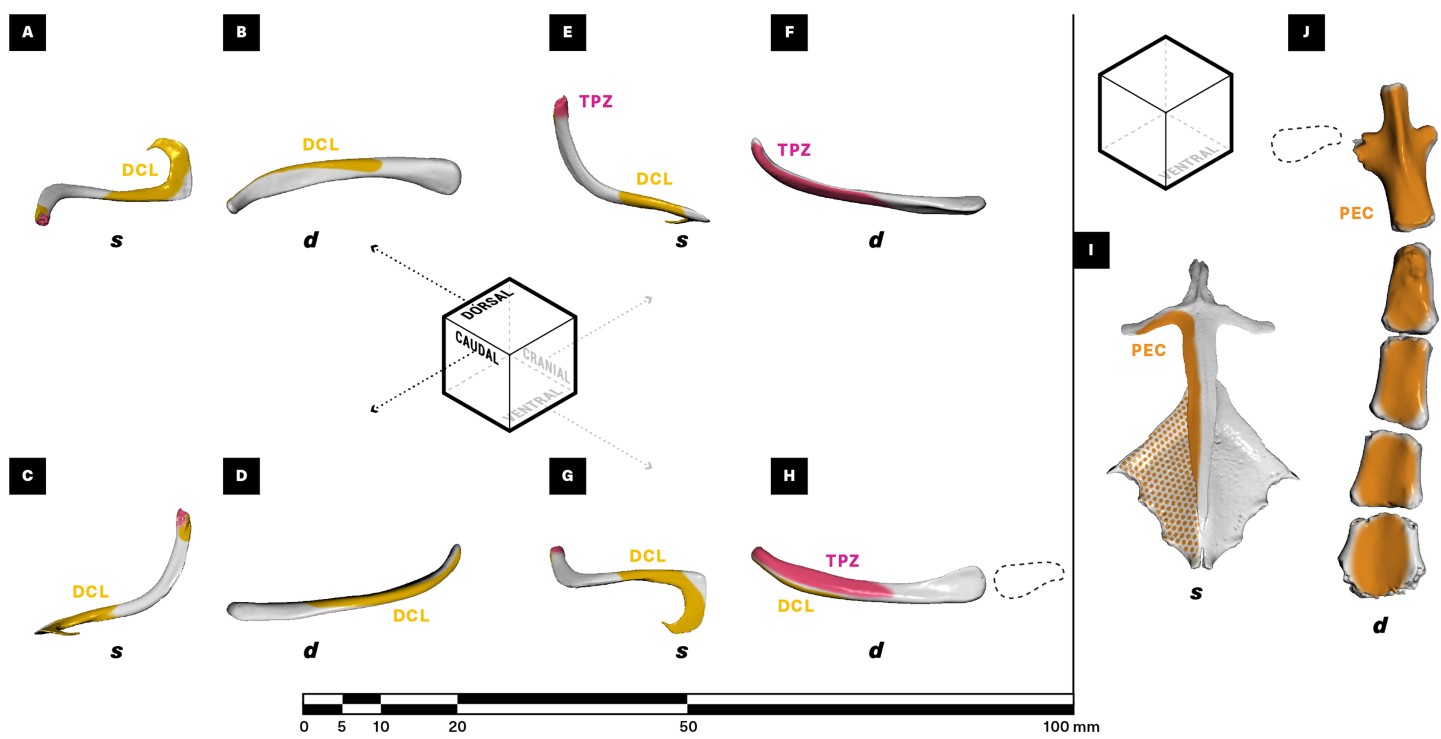

**Figure 5 Muscle attachments on the right clavicle (A–H), sternum + interclavicle (I), and sternum (J) of *Salvator merianae* (s) and *Didelphis virginiana* (d).** Clavicle shown in dorsal (A and B), caudal (C and D), cranial (E and F), and ventral (G and H) view; sternum + interclavicle and sternum shown in ventral view (I and J). Stippled areas represent loose fascial associations between muscle and bone. Dashed black outline represents cartilaginous element in sternoclavicular joint. Muscle abbreviations and color-coding follow Fig. 2.

applies to the ulna and radius; the bones of the pectoral girdle are oriented with respect to the animal's body axis.

## Skeletal observations

Physical dissection revealed that both the tegu and the opossum possess intra-girdle mobility. In the tegu, the coracoids (=metacoracoid sensu *Vickaryous & Hall, 2006*) translate craniocaudally along the sternum in a sliding, tongue-and-groove articulation, similar to the Savannah monitor *Varanus exanthemicus* (*Jenkins & Goslow, 1983*). Both the acromioclavicular joint and the clavo-interclavicular joint are able to rotate to accommodate this motion, forming an open-chain three-bar linkage. Ligaments act to constrain the extremes of mobility at all three joints. The cranioventral borders of the coracoid cartilages overlap asymmetrically, with the right coracoid lying dorsal to the left in all the individuals studied.

In the opossum, large gaps were observed in the μCT data between the clavicle and its lateral and medial articulations with the acromion and the manubrium, respectively. Physical dissection revealed radiolucent, cartilaginous elements in these regions. The medial element is particularly noteworthy: found interposed between the medial end of each clavicle and the clavicular notches of the manubrium, it is flattened and paddle-shaped, with a rounded lateral (clavicular) end and a tapered medial (manubrial)

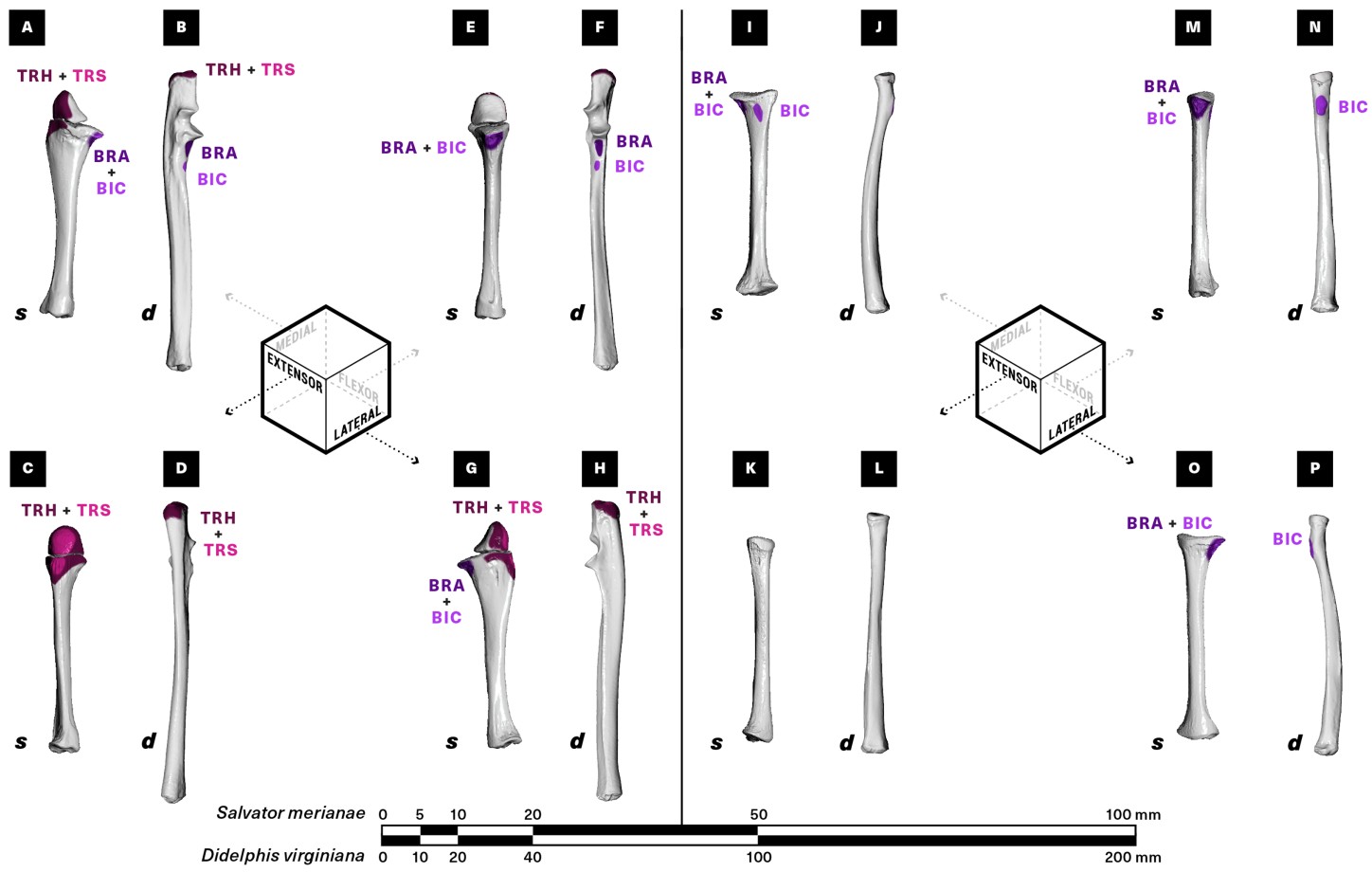

**Figure 6 Muscle attachments on the right ulna (A–H) and radius (I–P) of *Salvator merianae* (s) and *Didelphis virginiana* (d).** Shown in medial (A and B, I and J), extensor (C and D, K and L), flexor (E and F, M and N), and lateral (G and H, O and P) view. Muscle abbreviations and color-coding follow Fig. 2.

end (Fig. 5). This element appears to behave as a tensile link, as it buckles laterally when placed under compressive axial load. As with most therians, the intra-girdle mobility in the opossum occurs at the sternoclavicular and acromioclavicular joints, which provide almost unrestricted motion once the extrinsic muscles spanning the shoulder (e.g., m. pectoralis, m. latissimus dorsi, m. trapezius) are severed.

## Muscle observations

### M. latissimus dorsi (LAD; Figs. 2–4)

M. latissimus dorsi forms a large triangular sheet immediately deep to m. trapezius in both the tegu and the opossum (Fig. 2), originating proximally from the thoracodorsal/thoracolumbar fascia and converging to insert via a short, flat tendon onto the medial surface of the proximal humeral diaphysis (Fig. 4). A linear rugosity on the humerus marks the area of attachment in both taxa, stretching obliquely onto the extensor surface of the humerus in the tegu and running parallel to the diaphysis in the opossum. M. latissimus overlaps the caudal margin of the suprascapula in the tegu, where it is loosely

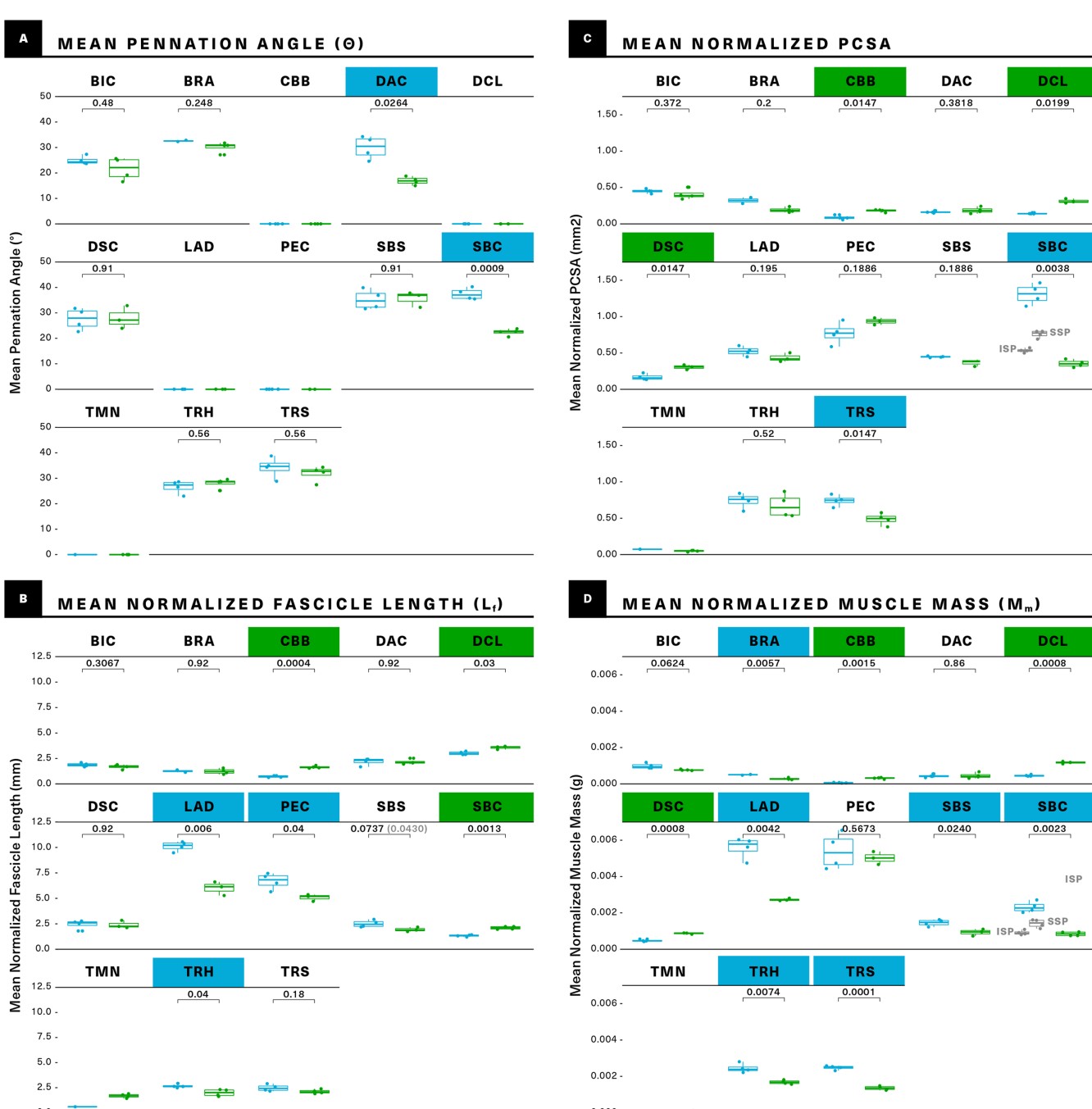

**Figure 7** **Statistical comparisons of muscle architectural properties between *Salvator merianae* (green) and *Didelphis virginiana* (blue) using unpaired two-sample Student's *t*-tests (α = 0.05).** (A) Mean pennation angle; (B) mean normalized fascicle length; (C) mean normalized PCSA; (D) mean normalized muscle mass. Green tiles indicate *Salvator* is significantly greater, while blue tiles indicate *Didelphis* is significantly greater; *white tiles* reflect no significant difference. PCSA and $M_m$ for *Didelphis* m. supraspinatus (SSP) and m. infraspinatus (ISP) are shown separately in gray, under m. supracoracoideus. M. infraspinatus and m. supraspinatus values for θ and $L_f$ are too close together for differences to be visible at this scale, and are not shown in the figure. *P*-values shown are adjusted for multiple comparisons using the Benjamini–Hochberg procedure (also given in Table S3). $L_f$ of m. subscapularis was significantly different prior to correction; uncorrected *P*-value shown in parentheses. Muscle abbreviations follow Fig. 2.

**Table 1 Muscle architecture parameters for the Argentine black and white tegu (*Salvator merianae*) and the Virginia opossum (*Didelphis virginiana*).**

*Salvator merianae*

| | $M_m$ (g) | $M_{mtu}$ (g) | $L_m$ (mm) | $L_{mtu}$ mm | $L_f$ (mm) | $\theta$ (°) | PCSA (mm²) |
|---|---|---|---|---|---|---|---|
| LAD | 2.73 ± 0.07 | 2.73 ± 0.07 | 93.51 ± 10.94 | 96.23 ± 11.63 | 60.12 ± 6.70 | 0 ± 0 | 43.45 ± 6.14 |
| PEC | 5.02 ± 0.36 | 5.02 ± 0.36 | 90.14 ± 7.23 | 90.14 ± 7.23 | 50.86 ± 3.42 | 0 ± 0 | 79.19 ± 4.66 |
| DAC | 0.45 ± 0.16 | 0.45 ± 0.16 | 26.19 ± 3.48 | 26.19 ± 3.48 | 21.57 ± 2.49 | 16.9 ± 1.6 | 18.66 ± 4.28 |
| DCL | 1.17 ± 0.07 | 1.17 ± 0.07 | 46.33 ± 3.31 | 46.33 ± 3.31 | 35.67 ± 1.69 | 0 ± 0 | 31.28 ± 3.08 |
| DSC | 0.87 ± 0.04 | 0.87 ± 0.04 | 37.85 ± 3.00 | 37.85 ± 3.00 | 24.10 ± 3.81 | 28.0 ± 4.5 | 30.49 ± 3.30 |
| SPC | 0.85 ± 0.11 | 0.85 ± 0.11 | 37.51 ± 1.66 | 37.51 ± 1.66 | 21.05 ± 1.43 | 22.4 ± 1.3 | 35.50 ± 5.13 |
| SHA = TMN | 0.10 ± 0.02 | 0.10 ± 0.01 | 19.75 ± 1.84 | 20.80 ± 2.67 | 18.44 ± 2.02 | 0 ± 0 | 4.94 ± 1.04 |
| SBS | 0.93 ± 0.19 | 0.93 ± 0.19 | 29.06 ± 3.68 | 29.06 ± 3.68 | 19.36 ± 2.18 | 35.6 ± 3.0 | 36.80 ± 4.84 |
| SBC | 0.70 ± 0.12 | 0.7 ± 0.12 | 29.79 ± 2.19 | 29.79 ± 2.19 | 23.04 ± 2.29 | 24.7 ± 2.9 | 26.13 ± 3.67 |
| CBB | 0.31 ± 0.04 | 0.31 ± 0.04 | 19.86 ± 2.69 | 19.86 ± 2.69 | 16.49 ± 1.24 | 0 ± 0 | 17.98 ± 1.86 |
| CBL | 0.40 ± 0.01 | 0.4 ± 0.01 | 40.17 ± 2.70 | 44.11 ± 3.22 | 26.70 ± 1.90 | 17.8 ± 2.8 | 13.71 ± 1.10 |
| BIC | 0.77 ± 0.03 | 0.81 ± 0.04 | 31.21 ± 2.51 | 46.11 ± 3.30 | 16.81 ± 2.17 | 21.6 ± 4.5 | 40.64 ± 6.93 |
| TRS | 1.36 ± 0.11 | 1.4 ± 0.09 | 35.41 ± 3.69 | 39.82 ± 4.74 | 22.67 ± 2.07 | 31.8 ± 3.0 | 48.70 ± 8.09 |
| TRH = TRM + TRL | 1.68 ± 0.11 | 1.7 ± 0.11 | 30.94 ± 4.54 | 35.48 ± 5.29 | 21.46 ± 3.37 | 28.0 ± 1.9 | 67.41 ± 16.10 |
| BRA | 0.28 ± 0.05 | 0.28 ± 0.05 | 30.11 ± 4.22 | 30.11 ± 4.22 | 12.30 ± 2.68 | 30.1 ± 2.0 | 19.13 ± 3.47 |

*Didelphis virginiana*

| | $M_m$ (g) | $M_{mtu}$ (g) | $L_m$ (mm) | $L_{mtu}$ mm | $L_f$ (mm) | $\theta$ (°) | PCSA (mm²) |
|---|---|---|---|---|---|---|---|
| LAD | 5.58 ± 0.59 | 5.68 ± 0.52 | 145.37 ± 7.25 | 152.73 ± 9.18 | 101.17 ± 4.78 | 0 ± 0 | 52.38 ± 6.46 |
| PEC | 5.40 ± 0.99 | 5.41 ± 0.98 | 114.76 ± 2.98 | 114.76 ± 2.98 | 66.89 ± 7.98 | 0 ± 0 | 77.06 ± 15.05 |
| DAC | 0.44 ± 0.09 | 0.46 ± 0.11 | 27.43 ± 3.80 | 29.32 ± 4.94 | 21.84 ± 3.53 | 30.0 ± 4.5 | 16.27 ± 1.43 |
| DCL | 0.46 ± 0.05 | 0.46 ± 0.05 | 33.40 ± 4.06 | 33.40 ± 4.06 | 30.20 ± 1.65 | 0 ± 0 | 14.27 ± 0.77 |
| DSC | 0.48 ± 0.06 | 0.50 ± 0.05 | 33.38 ± 3.70 | 35.96 ± 3.41 | 24.43 ± 4.41 | 27.6 ± 4.3 | 16.87 ± 4.14 |
| SSP | 1.41 ± 0.24 | 1.50 ± 0.25 | 39.18 ± 1.50 | 41.08 ± 1.02 | 14.01 ± 1.28 | 37.9 ± 3.8 | 75.15 ± 10.74 |
| ISP | 0.91 ± 0.13 | 1.04 ± 0.19 | 38.25 ± 2.99 | 40.41 ± 3.37 | 12.64 ± 0.78 | 37.0 ± 1.4 | 54.25 ± 6.23 |
| TMN = SHA | 0.06 ± NA | 0.06 ± NA | 13.24 ± NA | 13.24 ± NA | 7.76 ± NA | 0 ± NA | 7.32 ± NA |
| SBS | 1.45 ± 0.18 | 1.56 ± 0.17 | 38.46 ± 5.53 | 38.46 ± 5.53 | 25.12 ± 3.42 | 35.2 ± 3.9 | 44.7 ± 1.01 |
| TMJ | 0.85 ± 0.18 | 0.88 ± 0.18 | 46.33 ± 3.39 | 46.33 ± 3.39 | 29.12 ± 3.88 | 29.8 ± 1.1 | 23.99 ± 4.58 |
| CBB | 0.06 ± 0.02 | 0.07 ± 0.01 | 11.09 ± 0.40 | 16.58 ± 1.41 | 7.26 ± 0.97 | 0 ± 0 | 8.69 ± 2.84 |
| BIC | 0.99 ± 0.14 | 1.14 ± 0.13 | 39.48 ± 2.76 | 50.89 ± 2.78 | 18.76 ± 1.82 | 24.9 ± 1.7 | 45 ± 3 |
| TRS | 2.48 ± 0.11 | 2.73 ± 0.11 | 40.54 ± 2.56 | 45.22 ± 2.94 | 26.27 ± 3.30 | 34.2 ± 4.1 | 74.31 ± 7.63 |
| TRH = TRM + TRL | 2.45 ± 0.25 | 2.54 ± 0.26 | 37.68 ± 3.19 | 42.36 ± 4.06 | 28.10 ± 1.86 | 26.6 ± 2.6 | 74.05 ± 10.38 |
| BRA | 0.50 ± 0.03 | 0.54 ± 0.03 | 38.05 ± 3.50 | 38.05 ± 3.50 | 12.58 ± 1.64 | 32.6 ± 0.4 | 32.23 ± 5.85 |

**Note:**
Normalized species means are scaled to a 1 kg animal. Abbreviations and symbols: $M_m$, muscle mass; $M_{mtu}$, muscle-tendon unit mass; $L_m$, muscle length; $L_{mtu}$, muscle-tendon unit length; $L_f$, fascicle length; θ, pennation angle; PCSA, physiological cross-sectional area; NA, insufficient samples to calculate error. Muscle abbreviations follow Fig. 2.

anchored by fascia (Fig. 3). While m. latissimus also overlaps the caudal margin of the scapula in the opossum, there is no direct contact with the surface of the bone.

The muscle is about twice as massive as in the opossum as compared to the tegu, but its significantly greater $M_m$ is accompanied by a greater $L_f$, giving the two muscles comparable PCSAs (Fig. 7; Table 1).

### M. pectoralis (PEC; Figs. 2–5)

While we were able to visually observe two distinct divisions of m. pectoralis in the tegu and three in the opossum, we were unable to mechanically or digitally divide the muscle on a consistent basis in either taxon, and make the simplifying assumption of a single, unified pectoralis complex for the purposes of quantitative analysis and comparison.

In the tegu, m. pectoralis is a relatively thick, roughly triangular muscle originating from the interclavicle and costal cartilages, with an additional loose fascial adhesion to the ventral surface of the sternum (Figs. 2 and 5). Contralateral pectoralis muscles are separated by a thin gap at the midline. The opossum m. pectoralis is more trapezoidal in shape (Fig. 2), originating extensively from the ventral surface of the sternum (Fig. 5); contralateral muscles are joined completely at the midline. Although m. pectoralis inserts on the deltopectoral crest of the humerus in both taxa, the deltopectoral crest itself differs markedly in shape between the tegu and the opossum, coming to an acute, triangular apex in the former while extending at least halfway down the diaphysis in a pronounced, linear ridge in the latter (Fig. 4). As a result, m. pectoralis converges towards its insertion in the tegu, whereas in the opossum most of the superficial layer of fascicles is oriented perpendicular to the animal's sagittal plane.

*M. pectoralis* does not differ significantly in $M_m$ or PCSA between the two taxa, although we note the relatively high variance in the opossum $M_m$ data. While neither animal shows any meaningful m. pectoralis pennation, $L_f$ is significantly greater in the opossum (Fig. 7; Table 1).

### Mm. deltoideus acromialis, deltoideus clavicularis, and deltoideus scapularis (DAC, DCL, DSC; Figs. 2–5)

Both the tegu and the opossum possess a deltoideus complex comprising three heads: m. deltoideus scapularis, m. deltoideus clavicularis, and m. deltoideus acromialis (Fig. 2). M. deltoideus scapularis in the tegu originates directly from the suprascapula along a narrow arc, starting at the caudal margin of the suprascapula and ending halfway down the ventrally-projecting process connecting the cranial margin of the suprascapula to the coracoid (Fig. 3). The medial surface of the muscle is loosely anchored by fascia to the caudal half of the lateral suprascapular surface. From its origin, m. deltoideus scapularis extends ventrally and caudally as a fan-shaped sheet (Fig. 2) to insert together with m. deltoideus clavicularis on the apex of the humeral deltopectoral crest (Fig. 4). In the opossum, m. deltoideus scapularis originates on the ventral two-thirds of the caudal surface of the scapular spine (Fig. 3), and inserts along a defined ridge running between the greater tubercle of the humerus and the entepicondyle (Fig. 4).

In the tegu, m. deltoideus clavicularis originates from the ventral surface of the hook-shaped medial end of the clavicle (Fig. 5), and from the ligamentous sheet binding the clavicle to the interclavicle. This muscle extends cranially to wrap around the cranial edge of the clavicle, doubles back to attach to the dorsal surface of the clavicle as well (Fig. 2), and finally runs caudally to insert via a short common tendon with the scapular head on the apex of the humeral deltopectoral crest (Fig. 4). Fascicles appear to be continuous across the fold between the cranial and caudal ends of the muscle. In the opossum, m. deltoideus clavicularis originates along the cranial edge of the distal two-thirds of the clavicle (Fig. 5), and inserts with m. pectoralis along the entire length of the medial side of the humeral deltopectoral crest (Fig. 4). This muscle is completely overlapped by the deeper m. pectoralis.

M. deltoideus acromialis appears in the tegu as a wedge-shaped muscle lying just deep to m. deltoideus scapularis (Fig. 2), and originating from the ventral process of the suprascapula immediately ventral to the origin of m. deltoideus scapularis (Fig. 3). The origin of this muscle extends caudally, spanning the suture between the scapula and coracoid to end at the glenoid fossa. Distally, the muscle inserts adjacent and proximal to the common insertion of m. deltoideus clavicularis and m. deltoideus scapularis, on the cranial margin of the deltopectoral crest (Fig. 4). In the opossum, m. deltoideus acromialis originates from a small notch across the tip of the acromion (Fig. 3), and inserts all over the lateral surface of the humeral deltopectoral crest (Fig. 4). Mm. deltoideus acromialis and deltoideus clavicularis are closely associated by connecting fascia in both the tegu and the opossum.

On an architectural level, similarities and differences are evident between the tegu and opossum mm. deltoideus. M. deltoideus scapularis has significantly greater $M_m$ and PCSA in the tegu than in the opossum, but does not differ significantly in $L_f$ or $\theta$; m. deltoideus clavicularis is parallel-fibered in both animals, but has significantly greater $M_m$, $L_f$, and PCSA in the tegu than in the opossum; and m. deltoideus acromialis does not differ significantly in $M_m$, $L_f$ or PCSA between the tegu and the opossum, although the opossum m. deltoideus acromialis is significantly more pennate (Fig. 7; Table 1).

### M. supracoracoideus = Mm. infraspinatus, supraspinatus (SPC, ISP, SSP; Figs. 2–4)

The mammalian infraspinatus and supraspinatus muscles are widely recognized as the probable homologues of the non-mammalian supracoracoideus, on the basis of developmental fate mapping (*Romer, 1944*; *Cheng, 1955*) and their shared innervation by a supracoracoid/suprascapular nerve.

In the tegu, m. supracoracoideus occupies the entire lateral surface of the bony coracoid cranial to the glenoid fossa, as well as most of the lateral surface of the coracoid cartilage (Fig. 3). This muscle runs deep to the clavicular head of the deltoid, inserting by a short tendon on the humeral deltopectoral crest proximal to mm. deltoideus scapularis and deltoideus clavicularis, and ventral to m. deltoideus acromialis (Fig. 4). In the opossum, mm. infraspinatus and supraspinatus originate laterally on the scapula, on opposite sides of the scapular spine and within the infraspinous and supraspinous fossae

respectively (Fig. 3). The infraspinatus inserts via a short tendon on the lateral surface of the greater humeral tubercle, while the supraspinatus inserts via a slightly longer tendon on the cranial surface of the greater tubercle (Fig. 4).

The tegu supracoracoideus is unipennate, while the opossum infraspinatus and supraspinatus are multipennate with significantly shorter $L_f$. Mm. infraspinatus and supraspinatus in the opossum sum to a significantly greater total $M_m$ and PSCA than m. supracoracoideus in the tegu (Fig. 7; Table 1).

### M. scapulohumeralis anterior = m. teres minor (SHA, TMN; Figs. 2–4)

Here we follow *Romer (1944)* and *Cheng (1955)* in homologizing the m. scapulohumeralis anterior found in lizards with the m. teres minor of mammals, although other interpretations have been put forward (see "Discussion").

In the tegu, m. scapulohumeralis anterior originates on the lateral surface of the scapula, ventral to the origin of the lateral part of m. subscapularis and dorsal to the origin of m. deltoideus acromialis (Fig. 3). This muscle passes medial to the cranio-dorsal cruciate ligament to insert directly on the extensor surface of the humerus near the lesser tubercle, immediately proximal to the insertion of m. latissimus dorsi (Fig. 4).

M. teres minor is present in the opossum as a small muscle closely associated with m. infraspinatus, but separated by a fascial plane. Due to the small size of this muscle and its proximity to the infraspinatus, we were only able to isolate it cleanly enough to record architecture measurements in one individual. The teres minor originates on the lateral surface of the scapula between the origin of triceps scapularis and the glenoid fossa, and is innervated by a branch of the axillary nerve. It inserts via a short tendon on the lateral surface of the greater humeral tubercle, immediately distal to the insertion of m. infraspinatus.

While we were unable to collect sufficient teres minor data from the opossum for statistical comparison, this muscle was not found to be pennate in either species, and may have a slightly greater $L_f$ in the tegu.

### Mm. subcoracoideus, subscapularis (SBC, SBS; Figs. 2–4)

The medial surface of the scapula is occupied by the origin of a large, multipennate m. subscapularis in both the tegu and the opossum (Fig. 3). In both animals, m. subscapularis inserts via a short tendon on the lesser tubercle of the humerus (Fig. 4). While m. subscapularis is confined to the medial surface of the scapula in the opossum, the muscle's origin in the tegu is expanded to include to the medial, ventralmost surface of the suprascapula as well as the medial surface of its descending process. Caudally, the tegu m. subscapularis wraps around the axillary border of the scapula to also originate on a convex, triangular area on the lateral surface of that bone, dorsal to the origins of m. triceps scapularis and m. teres minor. In *Varanus*, a muscle resembling the lateral portion of the tegu subscapularis is identified as m. scapulohumeralis posterior (*Jenkins & Goslow, 1983*), which should be innervated by a branch of the axillary nerve (*Fürbringer, 1900*). In the tegu, both the medial and lateral portions of m. subscapularis are supplied by a single nerve (n. subscapularis) and are impossible to mechanically or digitally separate,

indicating they form a single muscle. M. subscapularis has significantly greater mass and fascicle length in the opossum than in the tegu, but PCSA and pennation are not significantly different between the two taxa (Fig. 7; Table 1).

M. subcoracoideus originates from the medial surface of the coracoid in the tegu, inserting separately on the lesser tubercle. Given the drastic reduction of the coracoid in the opossum, and the difference in muscle activation timing between m. subscapularis and m. subcoracoideus (m. subscapularis primarily during stance, m. subcoracoideus primarily during swing) (*Jenkins & Goslow, 1983*), we consider m. subcoracoideus to be lost in the opossum, rather than incorporated into the subscapularis mass. No architectural comparisons are therefore made here.

### M. teres major (TMJ; Figs. 2–4)

M. teres major is present only in the opossum, as a long-fibered, unipennate muscle originating on the caudal margin of the scapula (Fig. 3) and inserting mid-diaphysis on the medial surface of the humerus, adjacent and slightly distal to the insertion of m. latissimus dorsi (Fig. 4).

No m. teres major is evident in the tegu. Although such a muscle has been described for crocodilians (*Meers, 2003*; *Klinkhamer et al., 2017*), turtles (*Walker, 1973*), and the lizard *Uromastyx* (*Lecuru-Renous, 1968*), its area of origin in the tegu is occupied by a portion of m. latissimus dorsi instead (Fig. 3).

### Mm. coracobrachialis brevis, coracobrachialis longus (CBB, CBL; Figs. 2–4)

In the tegu, both m. coracobrachialis brevis and m. coracobrachialis longus originate on the lateral side of the coracoid, caudal to the origin of m. biceps brachii (Fig. 3). M. coracobrachialis brevis is a trapezoidal sheet inserting all over the proximal flexor surface of the humerus and running onto the diaphyseal ridge forming the medial base of the deltopectoral crest (Figs. 2 and 4). M. coracobrachialis longus runs down the humeral diaphysis with the median nerve and brachial artery, inserting around the entepicondylar foramen (Fig. 4).

In the opossum, m. coracobrachialis brevis appears as a flattened, teardrop-shaped muscle closely associated with the glenohumeral capsule. It originates with m. biceps brachii as a shared tendon coming from the tip of the coracoid process (Fig. 3), and inserts directly on a short, linear area on the medial surface of the humerus, just proximal to the insertion of m. teres major (Fig. 4). No m. coracobrachialis longus is evident; a neurovascular bundle comprising the median nerve and the brachial artery courses to the entepicondylar foramen in its place.

M. coracobrachialis brevis is not pennate in either animal studied, but has significantly greater $M_m$, $L_f$, and PCSA in the tegu than in the opossum (Fig. 7; Table 1). M. coracobrachialis longus is weakly pennate in the tegu, and absent in the opossum.

### M. biceps brachii (BIC; Figs. 2, 3, 6)

M. biceps brachii is described in *Sphenodon* and *Iguana* as having two proximodistally-staggered, closely-associated heads (*Fürbringer, 1900*; *Lecuru-Renous, 1968*; *Jenkins & Goslow, 1983*), but we consistently found only one in the tegu. Two heads were observed in

the opossum but were too integrated to reliably divide for analysis. We thus consider m. biceps brachii to be a single, fusiform muscle with a single origin and multiple insertions for the purposes of the present study.

In the tegu, m. biceps brachii takes origin as a broad, flat tendon just ventral to the coracoid facet of the glenoid fossa, sandwiched between m. supracoracoideus cranially and m. coracobrachialis brevis caudally (Fig. 3). In the opossum, m. biceps brachii shares a single origin with m. coracobrachialis brevis on the tip of the coracoid process. In both animals, the muscle runs medial to m. brachialis down the length of the humerus (Fig. 2), before inserting via short tendons on the flexor surfaces of both the radius and the ulna (Fig. 6). In the tegu, m. biceps brachii shares its ulnar tendon and one of its two radial tendons with m. brachialis, with a third tendon inserting by itself on the medial surface of the proximal radius. In the opossum, m. biceps brachii inserts onto the ulna and the radius as two separate tendons, with one tendon attaching just distal to the m. brachialis insertion on the flexor surface of the proximal ulna, and the other inserting all over the radial tuberosity on the flexor surface of the proximal radius.

The architectural properties of m. biceps brachii do not differ significantly between the two taxa (Fig. 7; Table 1).

### M. triceps brachii (TRS, TRM, TRL; Figs. 2–6)

M. triceps brachii is present in both the tegu and the opossum as a large muscle complex occupying the extensor surface of the humerus (Fig. 2). The triceps complex of both animals comprises a superficial fusiform portion originating on the scapula (m. triceps scapularis = m. triceps longus) (Fig. 3), and a wider deep portion originating on the humerus (m. triceps humeralis = mm. triceps medialis, triceps lateralis) (Fig. 4). While the deeper medial and lateral heads of m. triceps brachii are visually distinct and originate from clearly delineated areas of the humeral diaphysis, they interdigitate for most of their length and are impossible to consistently divide, and are grouped here as m. triceps humeralis. An additional coracoid head has been described in various lepidosaurs (*Sphenodon* (*Fürbringer, 1900*), *Iguana* (*Romer, 1922*; *Lecuru-Renous, 1968*), and *Varanus* (*Jenkins & Goslow, 1983*)), but is absent in the tegu.

In both animals, m. triceps scapularis originates via tendon on the axillary border of the scapula, immediately dorsal to the origin of m. teres minor and the glenoid fossa (Fig. 3). The area of origin is a small ellipsoid adjacent to the cranio-dorsal cruciate ligament in the tegu, and a long, narrow strip in the opossum. Triceps scapularis is fusiform in the tegu but flattened proximally in the opossum. The muscle's belly runs down the humeral diaphysis superficial to the triceps humeralis, inserting via a thick common tendon on the olecranon process of the ulna (Fig. 6).

M. triceps humeralis consists of a medial and a lateral head. Both originate directly on the extensor surface of the humeral diaphysis (Fig. 4). The origins of the two heads are of roughly equal area in the tegu, but in the opossum the lateral origin is larger than the medial origin. M. triceps humeralis encloses the extensor surface of the humeral diaphysis and distal epiphysis, inserting with triceps scapularis on the ulnar olecranon process via a common triceps tendon (Fig. 6).

In both taxa, m. triceps scapularis is bipennate and m. triceps humeralis is multipennate. Both the scapular and humeral divisions have significantly greater $M_m$ in the opossum (Fig. 6). Additionally, the opossum m. triceps scapularis has a significantly greater PCSA, whereas the opossum m. triceps humeralis has a significantly greater $L_f$.

### M. brachialis (BRA; Figs. 2, 4 and 6)

M. brachialis is similar in the tegu and the opossum, originating in both as a direct attachment along the lateral and flexor surfaces of the humeral diaphysis, although in the opossum the origin extends onto the humeral extensor surface as well (Fig. 4). The tegu m. brachialis inserts via two common tendons with m. biceps brachii on the flexor surfaces of the proximal ulna and the proximal radius (Fig. 6). In the opossum, the brachialis inserts by itself on the flexor surface of the ulnar coronoid process (Fig. 6). M. brachialis has significantly greater $M_m$ in the opossum, but the two animals do not differ significantly in $L_f$, pennation or PCSA.

### Muscle specialization and architectural design

Figure 8 (alternative high-contrast color coding in, Fig. S1) is a scatter plot of PCSA (normalized to body mass$^{2/3}$) against $L_f$ (normalized to body mass$^{1/3}$) of muscles shared between the tegu and the opossum. This functional morphospace visualizes a tradeoff between force production and working range, and allows relative muscle specialization to be compared between the two species (*Lieber, 2002*; *Eng et al., 2008*; *Allen et al., 2010*; *Dick & Clemente, 2016*; *Dickson & Pierce, 2019*). Homologous muscles are generally architecturally similar between the tegu and the opossum and occupy similar regions of the functional morphospace (Fig. 8). PCSA tends to differ more than $L_f$ between the two species, with muscles in the opossum tending to exhibit greater PCSAs than their tegu homologues. This trend is reversed for the mm. deltoideus and m. coracobrachialis brevis of the tegu, which both exhibit greater PCSA. *M. pectoralis* may also possess a greater PCSA in the tegu, although this difference was not significant due to high variance in the opossum data.

Five of the 13 muscles show statistically significant architectural divergence that may reflect shifts in locomotor function. (1) M. latissimus dorsi and m. pectoralis possess a significantly greater working range in the opossum, but force production is similar between the animals. (2) The mm. deltoideus appear to be slightly reduced in the opossum relative to the tegu: the scapular and clavicular heads have lower force capacity, and the clavicular head has a shorter working range. (3) The scapular head of m. triceps is capable of significantly greater force production in the opossum. (4) The opossum coracobrachialis complex is reduced compared to the tegu: m. coracobrachialis longus is absent, and m. coracobrachialis brevis has a significantly lower force capacity and working range. (5) The opossum equivalents of m. supracoracoideus are adapted for significantly greater force production at the expense of working range. This relationship holds true regardless of whether we compare the tegu supracoracoideus to the opossum mm. infraspinatus and supraspinatus individually, or if the latter two are pooled together as a single muscle.
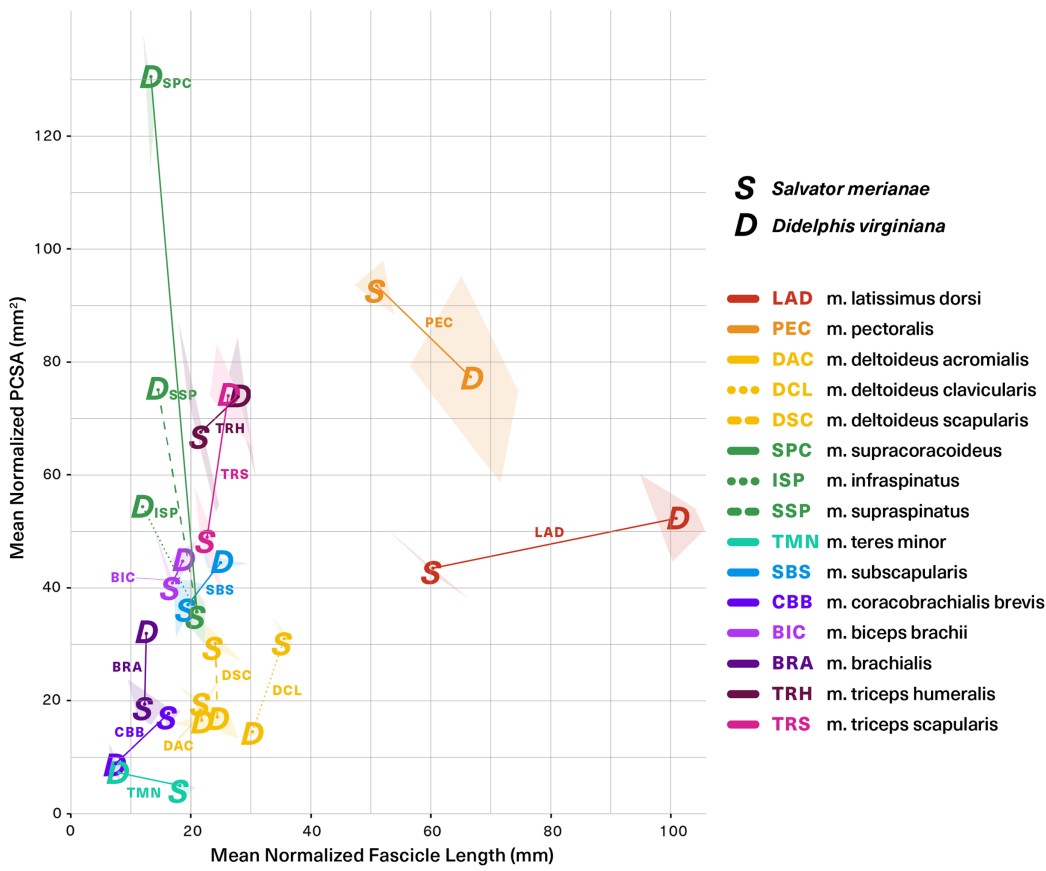

**Figure 8** **Functional morphospace comparing normalized PCSA against normalized fascicle length.** Muscles tend to vary along either one axis or the other, consistent with a tradeoff between force production ($y$-axis) and working range ($x$-axis). Muscle abbreviations and color-coding follow Fig. 2.

## Correcting for multiple comparisons

As the number of simultaneous statistical tests increases, so too does the chance of incorrectly rejecting the null (false positive/type I error), purely as a function of probability (*Pike, 2011*). It is standard practice to correct for this issue of multiple comparisons when evaluating a large number of hypotheses at the same time. Among the available correction methods, the Benjamini–Hochberg procedure controls the False Discovery Rate (FDR), or the proportion of incorrectly-rejected nulls (*Benjamini & Hochberg, 1995*). It is more powerful than the commonly-used Bonferroni and Holm corrections, and is considered more appropriate for situations where the number of hypotheses being tested is large relative to the number of samples (*Horn & Dunnett, 2004*). In the uncorrected analysis, 2/13 muscles differ significantly in $\theta$, 5/13 in PCSA, 7/13 in $L_f$, and 9/13 in $M_m$. After applying the Benjamini–Hochberg correction, m. subscapularis no longer differs significantly in $L_f$ between the tegu and the opossum, but all other differences remain significant at $\alpha = 0.05$.

## DISCUSSION

The pectoral locomotor apparatus was an important locus of anatomical and functional reorganization in the evolution of mammals. Morphological change during the transition to mammal-like posture and locomotion is documented by an extensive fossil record of non-mammalian synapsids (*Kemp, 2005*), raising the possibility of reconstructing in vivo musculoskeletal function via integrating fossilized skeletal morphology with inferred muscle anatomy. This study aimed to establish a framework for reconstructing non-mammalian synapsid musculature, and did so by comparing and contrasting the topology and architecture of muscles crossing the shoulder in an extant phylogenetic and morpho-functional bracket: the Argentine black and white tegu *S. merianae* and the Virginia opossum *D. virginiana*. Below, we first discuss the myological similarities between the two, which were numerous and striking. We evaluate conservation and convergence as two possible explanations for the similarities in functional design, and the broader implications for understanding amniote musculoskeletal evolution. We then detail instances where departures were observed, and interpret these differences in the light of known postural and locomotor contrasts. Finally, we demonstrate the applicability of our data towards estimating key input parameters in musculoskeletal modeling by inferring shoulder muscle PCSAs for an extinct non-mammalian synapsid. Future studies will build on this technique to investigate the evolution of posture and locomotion across the synapsid-mammal transition.

### Broad similarities in muscle topology and architecture

Pectoral girdle musculature is markedly similar between the tegu and the opossum. Based on physical and digital dissection, we found strong evidence for anatomical homology in 13 out of 18 muscles across the two species, including resurrecting a formerly-dismissed correspondence between the lizard m. scapulohumeralis anterior and the therian m. teres minor (Text S1). Muscle topology follows skeletal morphology: while the proximal shoulder muscle attachments have migrated with the modification of the pectoral girdle bones in the opossum, they remain recognizable between the two taxa; further, muscle attachments are directly comparable on the long bones of the forelimb, which differ little between the tegu and the opossum. These findings bolster a growing body of evidence from the extant phylogenetic bracket for a common set of amniote pectoral girdle and shoulder muscles (*Abdala & Diogo, 2010*; *Lai, Biewener & Pierce, 2018*), while adding new architectural data for these structures.

With our data we can go beyond gross morphology and ask: does internal architecture parallel topological similarity between muscle homologues? We find that, once scaled to body mass, the architectural similarities between homologous muscles outnumber the differences (Fig. 7). Muscle mass is significantly different in many cases, but is frequently accompanied by significant differences in fascicle length, resulting in fewer instances of significantly different PCSA. Mean pennation angle is very similar, differing significantly between the tegu and the opossum in only a couple of cases. The fact that PCSA and pennation are more similar between the two species than muscle mass and fascicle length suggests that differing aspects of muscle function might be under differing selective

pressures. Muscle PCSA relates to the force requirements for body support and locomotion (*Eng et al., 2008*). Pennation is linked to PCSA and force capacity for a given volume of muscle, but may also enable dynamic gearing of mechanical output (*Azizi, Brainerd & Roberts, 2008*). These functional traits almost certainly interact with one another within the confines of the shoulder and forelimb, and a balance between competing functional priorities may explain the architectural similarity observed between the morphologically-conservative amniotes studied here.

A possible explanation for the broad parallels shown here is that both the tegu and the opossum have retained many elements of the amniote last common ancestor's (LCA) pectoral musculoskeletal system largely unmodified. These affinities echo the findings of prior comparative work, in particular that of Jenkins and colleagues (*Jenkins & Weijs, 1979*; *Jenkins & Goslow, 1983*), who compared muscle gross anatomy and function between the Savannah monitor *Varanus exanthemicus* and the Virginia opossum, hypothesizing a set of "functional equivalences" between lizard and therian shoulder muscles based on similarities in muscle activation patterns. Another possibility is that these myological characteristics evolved convergently, and are simply representative of a smaller-bodied terrestrial generalist phenotype. Both conservation and convergence have their issues: conservation is made less likely by the 320 million-plus years since the amniote LCA (*Benton, 2009*), while convergence is difficult to assess without prior knowledge of plesiomorphic states in the sauropsid and synapsid lineages. More extensive collection of architectural data from other extant generalist amniotes would be helpful in determining between conservation and convergence: *Sphenodon* is an early-diverging lepidosaur that attains comparable adult body sizes to *S. merianae* and *D. virginiana* (*Feldman et al., 2016*), and would be a useful point of comparison in inferring the plesiomorphic lepidosaur condition. Larger varanid lizards could provide a perspective from a different extant locomotor generalist; while the pectoral limb anatomy (*Jenkins & Goslow, 1983*) and pelvic limb architecture (*Dick & Clemente, 2016*) of varanids are well-studied, the architecture of the pectoral limb has not been published on. Among extant therians, quolls, tasmanian devils, hyraxes, and some of the more terrestrial civets are all locomotor generalists of similar size whose muscle architecture has yet to be studied. Finally, the morphologically-specialized (*Jenkins, 1971a*) extant monotremes invite comparison as similarly-sized phylogenetic intermediates between tegus and opossums. While a musculoskeletal model for the short-beaked echidna *Tachyglossus aculeatus* already exists (*Regnault & Pierce, 2018*), our ability to interpret this animal's joint mobility and muscle moment arms will improve once its muscle architecture is described as well.

## Anatomical differences reflect locomotor transformation

Of the five muscles not directly shared between the tegu and the opossum, all have known origins in the amniote pectoral musculoskeletal system. M. subcoracoideus and m. coracobrachialis longus are probably plesiomorphic for amniotes and secondarily lost in therians; they are present in amphibians and monotremes as well as the tegu, but absent in the opossum (*Walthall & Ashley-Ross, 2006*; *Diogo et al., 2009*; *Gambaryan et al., 2015*). The remaining three muscles are found in the opossum and not the tegu, but are not

neomorphic. M. supraspinatus and m. infraspinatus have been shown to be differentiated developmental homologues of the lizard m. supracoracoideus (*Romer, 1944*; *Cheng, 1955*). Meanwhile, m. teres major is found in mammals, crocodile-line archosaurs, turtles, and at least one lizard, and is suggested to be an amniote character present in the LCA and secondarily lost in bird-line archosaurs as well as most lepidosaurs (*Abdala & Diogo, 2010*). Both of the muscles that are lost in therians (m. subcoracoideus and m. coracobrachialis longus) originate on the procoracoid and metacoracoid in the amniotes that possess them. Meanwhile, m. supraspinatus, m. infraspinatus, and m. teres major all take origin on the scapula, which in therians has uniquely expanded to comprise almost the entirety of the pectoral girdle. The synapsid fossil record preserves a marked trend towards coracoid reduction and scapular expansion, extending well into crown mammals (*Romer, 1922*; *Romer & Price, 1940*; *Jenkins, 1971a*; *Luo, 2015*). The shifting proportions of these skeletal elements and the accompanying dimensional and positional changes to the associated muscles are likely strongly linked to postural and locomotor evolution in synapsids.

Comparative study of architecture has the potential to reveal aspects of locomotor specialization (*Allen et al., 2014*; *Böhmer et al., 2018*). Hence, architectural differences in muscles that are shared between the tegu and the opossum may also be interpreted in light of the postural and locomotor differences between these animals. The longer fascicles of the opossum's m. latissimus and m. pectoralis give these muscles a greater working range, which may accommodate the longer strides and greater pectoral girdle mobility of therians (*Sereno & McKenna, 1995*). The smaller PCSAs of the opossum's clavicular and scapular deltoids compared to the tegu may signify a decreased reliance on these muscles for limb protraction during swing phase, since girdle mobility may be a greater factor than glenohumeral joint mobility in the therian stride compared to other amniotes (*Fischer et al., 2002*; *Baier & Gatesy, 2013*). The unusual, folded morphology of the tegu's clavicular deltoid may work in conjunction with longer parallel-fibered fascicles to increase the muscle's working range beyond what would be achievable with a conventional parallel-fibered muscle, and may again reflect a greater role in humeral protraction for the deltoids of a "sprawling" animal.

The triceps complex is notably more massive in the opossum, but its architecture suggests different functional specializations between the scapular and humeral heads. The opossum m. triceps scapularis exhibits a ~50% increase in PCSA over its tegu counterpart but has a similar mean fascicle length. By comparison, the two humeral heads of the triceps have the same PCSA in both animals, but with significantly longer fascicles in the opossum. The increased PCSA of the opossum m. triceps scapularis suggests adaptation to resist larger ground reaction flexor moments at the elbow, as a result of shifting to an erect forelimb posture. Meanwhile, the longer fascicles of the opossum m. triceps humeralis likely provide a wider working range for the muscle. Taken together, these features provide evidence for a functional differentiation within the triceps complex, with the scapular head becoming specialized for force control in therians, and the humeral heads becoming specialized for position control of the zeugopod.

Both m. supracoracoideus and its derivatives m. supraspinatus and m. infraspinatus are thought to stabilize the glenohumeral joint during locomotion (*Jenkins & Weijs, 1979*; *Jenkins & Goslow, 1983*). In therians, a rotator cuff comprising m. supraspinatus, m. infraspinatus, m. subscapularis, and m. teres minor pulls on the humerus from opposing directions. The resulting compression of the humeral head against the glenoid fossa has been shown to be an important source of dynamic stability in ball-and-socket glenohumeral joints (*Lippitt & Matsen, 1993*; *Hsu et al., 2011*), and is thought to be particularly important in intermediate poses of the shoulder when the capsule and ligaments are relaxed and unable to contribute to joint stability. The greater PCSAs of the m. supraspinatus and m. infraspinatus could thus serve to stabilize the humerus as it undergoes the rapid, rhythmic oscillations generated by the highly-tuned therian neuromuscular system (*Jenkins & Goslow, 1983*; *Ross et al., 2013*). By comparison, the smaller PCSA and longer fascicles of m. supracoracoideus may reflect a greater reliance by other amniotes on ligaments rather than muscle force (*Haines, 1952*) to stabilize the shoulder, and may facilitate protraction of the humerus as it moves through the long, horizontal arcs typical of "sprawling" locomotion (*Jenkins & Goslow, 1983*; *Baier & Gatesy, 2013*).

## Reconstructing synapsid musculoskeletal evolution

Whether conservation or convergence is ultimately responsible for the architectural similarities found here, it is apparent that selective pressures have either maintained or resulted in a distinct terrestrial generalist architectural phenotype shared by the tegu and the opossum. As non-mammalian synapsids are typically reconstructed as terrestrial generalists, the architectural parameters of the tegu and opossum may provide realistic "bookends" for the phylogenetically- and morphologically-intermediate non-mammalian synapsids. Here, we illustrate this by estimating shoulder musculature PCSA in *Massetognathus pascuali*, a Triassic traversodontid cynodont for which a phylogenetically-bracketed musculoskeletal reconstruction has been published (*Lai, Biewener & Pierce, 2018*). Using a scaling equation to predict total body mass based on the circumferences of the humeral and femoral diaphyses (*Campione & Evans, 2012*), we estimate the body mass of a particular *Massetognathus pascuali* individual (MCZVP 3691) as 1.437 kg, with upper and lower bounds given by mean percent prediction error (PPE) as 1.806 kg and 1.069 kg respectively (Table S4). This prediction closely matches the tegus (1.33 ± 0.11 kg) and opossums (1.35 ± 0.26 kg) dissected in the present study.

Assuming geometric similarity, we can scale mass-normalized PCSAs from both extant animals by *Massetognathus'* predicted body mass$^{2/3}$ to arrive at architectural estimates reflecting contrasting postural paradigms (Fig. 9). Error around estimated PCSA is calculated by scaling by the upper and lower bounds of the body mass prediction. By taking the mean of the tegu and opossum values, we can obtain a third, intermediate estimate that might better represent an extinct transitional form. As expected from the broad similarities in PCSA previously discussed, *Salvator*-like, *Didelphis*-like, and intermediate values tend to fall fairly close to one another. In seven of the 16 muscles reconstructed both the *Salvator*-like and the *Didelphis*-like estimate are included within the upper and

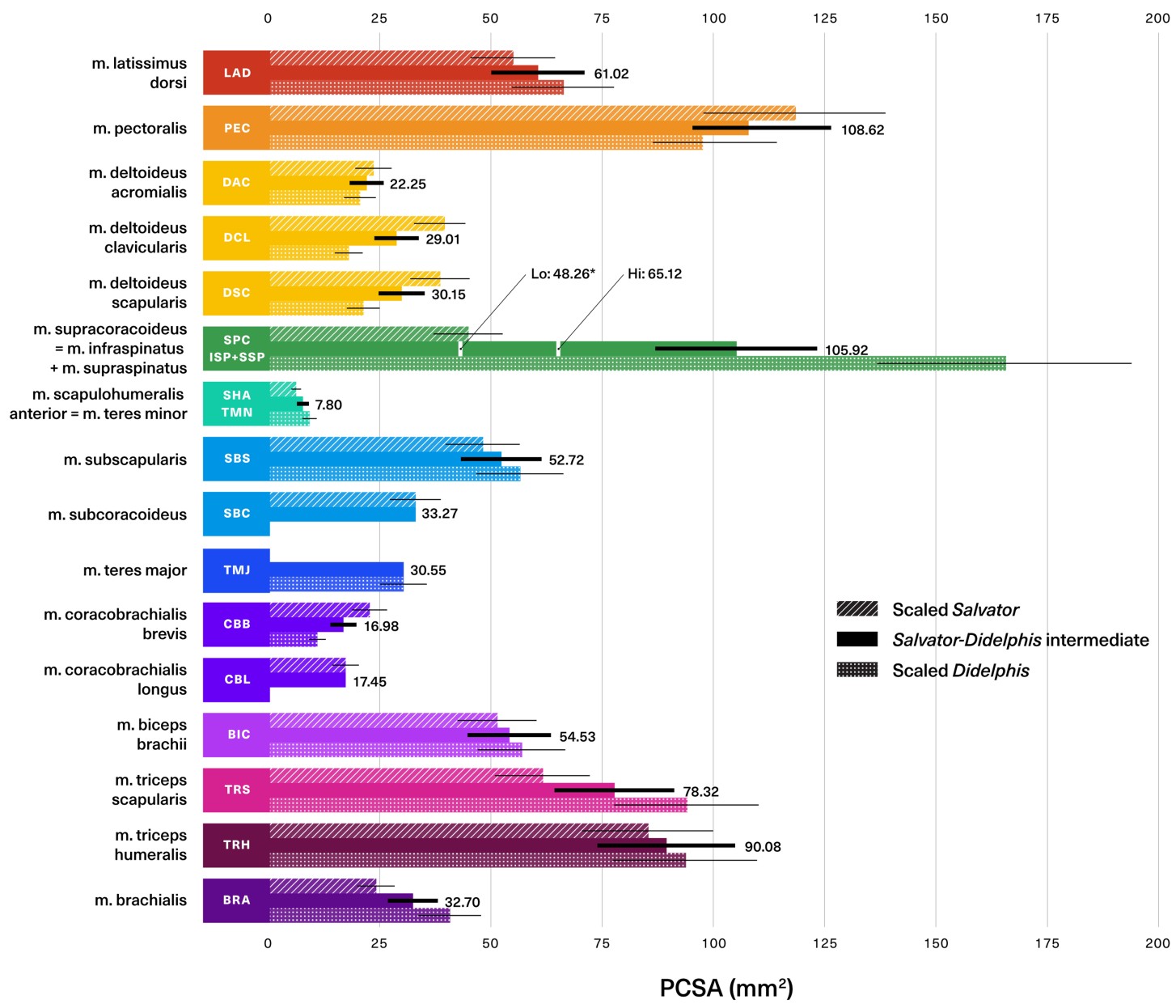

**Figure 9 Predicted muscle physiological cross-sectional areas (PCSAs) for traversodontid cynodont *Massetognathus pascuali* (MCZVP 3691).** *Salvator*-like, *Didelphis*-like, and intermediate reconstructions are depicted with diagonally-hatched, dotted, and solid bars, respectively. Predictions are based on a reconstructed body mass of 1.437 kg. Black numbers give the value of the intermediate estimate for each muscle. Black lines show uncertainty around predictions, calculated by scaling PCSAs using the upper and lower end estimates of *Massetognathus'* body mass (from percent prediction error (PPE) reported by *Campione & Evans (2012)*). SBC, TMJ and CBL lack error bars, as they are reconstructed from only one side of the bracket. Morphologically-informed low- and high-end estimates for the *Massetognathus* m. supracoracoideus are labeled as "Lo" and "Hi" respectively (see Table S5). Muscle abbreviations and color-coding follow Fig. 2. MCZVP: Museum of Comparative Zoology, Vertebrate Paleontology.

lower bounds of the intermediate estimate (thick black lines, Fig. 9); they fall just outside the bounds of the intermediate estimate in another five muscles. Altogether, this suggests that discrepancies between *Salvator*-like and *Didelphis*-like estimates of PCSA are generally proportionately small, and tend to be either closely comparable to or dominated

by uncertainty around estimated body mass. This provides confidence in the intermediate estimate, and at the same time illustrates the importance of conducting sensitivity testing in future work, so as to ascertain the relative contributions of the bracket and predicted body mass towards uncertainty in muscle parameter estimation.

While the *Salvator-Didelphis* intermediate seems to provide a reasonable estimate for the bulk of the shoulder musculature, a different approach is clearly needed for the m. supracoracoideus/mm. infraspinatus + supraspinatus group, where the opossum-like estimate exceeds the tegu-like estimate by a factor of approximately four (Fig. 9). The fourfold difference in normalized PCSA reflects the smaller relative size of this muscle in the tegu (0.85 g) compared to the opossum (2.32 g). However, as shown in the Table S5, the normalized ratio of PCSA: muscle origin area is similar between the tegu m. supracoracoideus (0.18) and its homologues mm. infraspinatus + supraspinatus in the opossum (0.15). Moreover, mm. infraspinatus and supraspinatus PCSAs have been shown to covary with the size of their areas of origin in the short-nosed bandicoot *Isoodon fusciventer*, a marsupial with pectoral skeletal anatomy comparable to *Didelphis* (*Martin et al., 2019*). Considering mm. supracoracoideus, infraspinatus, and supraspinatus are all flat, broad, pennate muscles with fleshy origins that occupy well-delineated areas on the scapulocoracoid or scapula (Fig. 3), relative origin area may be a reasonable proxy for estimating PCSA in *Massetognathus*. *Lai, Biewener & Pierce (2018)* reconstructed *Massetognathus* with an m. infraspinatus on the infraspinous fossa of the scapula, as well as an m. supracoracoideus/incipient m. supraspinatus on the anterior coracoid (procoracoid sensu *Vickaryous & Hall, 2006*) (Fig. S2B). The area of origin for m. infraspinatus in *Massetognathus* resembles that of the same muscle in the opossum (Figs. S2B–S2C), whereas the area of origin for the incipient m. supraspinatus in *Massetognathus* resembles that of the tegu m. supracoracoideus (Figs. S2A–S2B). Taking the PCSA: muscle origin area ratios for the opossum m. infraspinatus and the tegu m. supracoracoideus, we are able to reconstruct the PCSAs of mm. infraspinatus and incipient supraspinatus in a 1.437 kg *Massetognathus* as 29.40 mm$^2$ and 18.86 mm$^2$ respectively, resulting in a combined PCSA of 48.26 mm$^2$. This is a plausible estimate for *Massetognathus*, falling in between the PCSAs of the tegu (35.50 mm$^2$) and the opossum (130.84 mm$^2$). An alternative, higher estimate (65.12 mm$^2$) using the proportions of the tegu m. supracoracoideus and the opossum m. supraspinatus for the mm. infraspinatus and supraspinatus (respectively) of *Massetognathus* also falls between the tegu and the opossum.

We note several caveats for reconstructing non-mammalian synapsid myology using this approach. While evidence exists that PCSA and L$_f$ for many limb muscles scale at or close to isometry in varanid lizards (*Dick & Clemente, 2016*), proximal limb muscles scale with positive allometry in extant mammals and certain crocodiles (though not alligators) (*Alexander et al., 1981*; *Allen et al., 2014*). Not all non-mammalian synapsids were comparable in size to tegus and opossums; many, such as the Permian caseids and the Permo–Triassic dicynodonts, may have been up to three orders of magnitude larger (*Reisz & Fröbisch, 2014*; *Sulej & Niedźwiedzki, 2019*; *Romano & Manucci, 2019*), and architectural allometry may have to be taken into account. More studies of architectural allometry within extant squamates and therians are needed to determine the salience of

body size to muscle functional design. Further, the ultimate mechanical consequences (e.g., for muscle force, strain, and work production) are difficult to determine without modeling the musculoskeletal system as a whole. With that being said, general agreement among most of these estimates of PCSA provides confidence that fossil muscle architecture can be empirically inferred within reasonable limits, and the robustness of parameter estimates for reconstructing locomotor evolution can be assessed in future work by performing sensitivity testing on musculoskeletal models incorporating inferred architecture.

## CONCLUSION

Here, we compared shoulder myology across a phylogenetic and morpho-functional bracket for non-mammalian synapsids, consisting of the Argentine black and white tegu and the Virginia opossum. Our data revealed broad topological and architectural similarities between the tegu and the opossum, suggesting either conservation of plesiomorphic amniote myology or convergence towards a similar terrestrial generalist phenotype. In particular, muscle attachments on the humerus were directly comparable between both species, as were PCSAs and pennation across most muscles. The few topological and architectural differences can be interpreted in terms of functional tradeoffs associated with reduction of the mammalian shoulder girdle and the evolution of upright limb posture. For instance, we found significantly-increased PCSAs and shorter fascicles in the opossum m. infraspinatus and m. supraspinatus relative to their reptilian homologue (m. supracoracoideus), consistent with their enlarged scapular attachments and their role in stabilizing the therian ball-and-socket glenohumeral joint. Similarly, the opossum m. pectoralis and m. latissimus dorsi, which span the axial skeleton and the humerus, possess elongated fascicles that accommodate the increased mobility of the therian pectoral girdle. Both the myological similarities and differences are informative in reconstructing unpreserved muscle parameters in fossil synapsids, as we illustrate with the traversodontid cynodont *Massetognathus pascuali*. This work establishes the first quantitative basis for inferring functionally-important features of muscle architecture in extinct non-mammalian synapsids, and represents a critical first step in understanding how musculoskeletal reorganization led to the evolution of the versatile mammalian forelimb, with its myriad functions and behaviors.

## ACKNOWLEDGEMENTS

We thank Emma Hanslowe, Jillian Josimovich, Bryan Falk, and Robert Reed at the United States Geological Survey Daniel Beard Center for providing the tegu cadavers used in this study, and Tom French at the Massachusetts Division of Fisheries and Wildlife for supplying us with the opossum cadavers. Jessica Cundiff, José Rosado, and Joe Martinez assisted with specimens and materials in the Museum of Comparative Zoology. The Pierce and Biewener labs provided valuable feedback, in particular Katrina Jones and Sophie Regnault. We are grateful to Editor Virginia Abdala, as well as the reviewers Karl Bates, Andrew Cuff, and Miriam Mariana Morales, whose thoughtful comments were instrumental in improving this manuscript.

### Funding

This project was supported by National Science Foundation Grant No. DEB-1754459 (Stephanie E. Pierce), the Department of Organismic and Evolutionary Biology at Harvard University (Philip Fahn-Lai), and the Robert A. Chapman Fellowship (Philip Fahn-Lai). Publication costs were supported by the Wetmore Colles fund (Philip Fahn-Lai). The funders had no role in study design, data collection and analysis, decision to publish, or preparation of the manuscript.

### Grant Disclosures

The following grant information was disclosed by the authors:
National Science Foundation: DEB-1754459.
Department of Organismic and Evolutionary Biology at Harvard University.
Robert A. Chapman Fellowship.
Wetmore Colles fund.

### Competing Interests

Stephanie E. Pierce is an Academic Editor for PeerJ.

### Author Contributions

- Philip Fahn-Lai conceived and designed the experiments, performed the experiments, analyzed the data, prepared figures and/or tables, authored or reviewed drafts of the paper, and approved the final draft.
- Andrew A. Biewener conceived and designed the experiments, authored or reviewed drafts of the paper, and approved the final draft.
- Stephanie E. Pierce conceived and designed the experiments, analyzed the data, authored or reviewed drafts of the paper, and approved the final draft.

### Data Availability

Raw muscle architecture measurements are available in the Supplemental Files.
The Raw CT Data are available at Morphosource: S25440, S25439. URL: https://www.morphosource.org/Detail/ProjectDetail/Show/project_id/847.

### Supplemental Information

Supplemental information for this article can be found online at http://dx.doi.org/10.7717/peerj.8556#supplemental-information.

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
