# Peer review of "Broad similarities in shoulder muscle architecture and organization across two amniotes: implications for reconstructing non-mammalian synapsids"

_PeerJ, doi:10.7717/peerj.8556_

## Round 0.1 · original submission · Minor Revisions

I have received three reviews of your work, all of them very positive. The reviewers advise minor revisions and I agree. However, as you will see, several of their observations are not minor at all. The reviewers did an excellent job of analyzing your manuscript and I would like you to take all their comments into full consideration. I have a few suggestions on my own. I would like you to better explain the selection of the taxa for your study. Maybe you can consider the paper by Montero et al. (2004): Atlas de Tupinambis rufescens (Squamata: Teiidae). Anatomia externa, osteología y bibliografía (https://www.researchgate.net/publication/301691860_Atlas_de_Tupinambis_rufescens_-_Archivos_PowerPoint). I agree with our reviewer #1 in that muscle similarities between Salvator and Didelphis are far from unexpected. In fact, conservative muscular morphology seems to be the rule rather than the exception among tetrapods. In any case, I am sure that you will find the reviews useful to improve your work.

·

Basic reporting

No problems in my opinion.

Experimental design

No problems in my opinion

Validity of the findings

No problems in my opinion

Additional comments

I really enjoyed reading this study. It deals with two fundamental and inter-connected issues in comparative anatomy and evolutionary biomechanics: (1) how different/adapted are muscles in animals that function in different ways, and subsequently (2) how should we reconstruct muscles in extinct taxa in order to understand how they functioned? In my view these are the most pivotal questions currently facing the field of evolutionary biomechanics. I think the approaches used are entirely appropriate and the research has been carried out to a high standard. I have very few concerns and I’m excited to see the paper published. Below I have listed some general thoughts about the paper, along with some minor points about specific sentences/issues. None represent major concerns and could be implemented as and where the authors think they will improve the manuscript.

General Comments
1. The one thing I felt the paper was lacking in slightly was a standalone appreciation of how different the tegu and opossum are in their gross skeletal anatomy and the way they walk. Then subsequently how different they are/might be to the transitional fossils like Massetognathus. Appropriate papers are referenced, but if, for example, I want to gauge how sprawling vs upright the tegu and opossum are then I have to piece that together by comparing several published papers. I don’t think this issue is trivial because the paper builds to a situation where you average the morphology of the tegu and opossum and apply that average muscle morphology to the fossil. So it would be directly helpful to anyone wanting to judge the choice of taxa and validity of the approach generally to be able to easily judge the morphological and functional differences of the tegu and opossum. Would it be possible to add a multi-part figure that summarises the 3D skeletal anatomy and locomotor dynamics of these two animals? For example, part A could show the 3D pectoral girdles and forelimbs of the tegu and opossum in mid-stance or neutral standing postures (possibly with Massetognathus included too?) and then in subsequent parts provide some graphed 3D kinematics of the two species overlaid on each other, utilising data from published studies?

2. Following on directly from the point above, the general ‘feeling’ that I think the paper leaves the reader with is a very positive one: (1) the tegu and opossum are excellent bracketing taxa for transitional fossils; (2) their muscles are “strikingly” similar; (3) and as a combined result of (1) and (2) we’re in an excellent position to quantitatively reconstruct muscle properties in transitional fossils. I have made some points below about whether the picture being painted is a little too positive, but regardless of that, my previous point is that more background information would help the reader judge this pyramidal/hierarchical argument. However, a better understanding within the paper of the scale of differences between the chosen extant taxa, and the extant and extinct taxa, would also help in terms of the wider applicability of this approach to other groups and evolutionary transitions. The generally positive picture that comes through here is a bit of contrast to dinosaur and hominid studies where the extant bracketing taxa have quite different muscles and this poses issues when viably reconstructing muscles in extinct taxa. Do we have a fundamentally different situation here with early mammals? Might the approach used herein (e.g. applying a mean of two extant species to extinct taxa) work in some instances but not others? I don’t think it’s essential to add this to the discussion section, but it might make the paper more interesting and widen its appeal.

3. Discussion, Lines 624-628: I couldn’t decide whether to discuss this here or under minor comments below, but opted for here because addressing this concern could be integrated into the broader discussion I’ve suggested above (if the authors want to). All estimates of body mass in extinct taxa inevitably come with uncertainty. In their original paper, Campione and Evans (2012) included confidence or prediction intervals with their mass equation and encouraged (and still encourage; Campione 2017, Palebiology) people to use them when presenting a mass estimate for an extinct animal. There’s been quite a bit written the issue since (Bates et al. 2015, Biology Letters; Campione 2017, Paleobiology) in the context of dinosaur body mass, but here the uncertainty of the mass prediction is directly relevant because you use the derived body mass to scale your muscle properties. I think some discussion of this issue is warranted. Maybe even some calculations/sensitivity analysis could be done? Also, if you have alternative ideas, then perhaps methods of deriving quantitative muscle properties that don’t require an estimated body mass could be discussed? Also, double check that you are using the correct measurements as inputs into the Campione & Evans (2012) equation. In your supplementary table you have labelled the columns “Mid-diaphyseal Circumference”, but I think that the equation is based on minimum shaft circumference (which might not necessarily be mid-shaft, unless that’s your definition of mid-shaft).

4. Five muscles are not directly shared between your two studied species. You could very easily quantify what impact this has on gross functional parameters. For example, how much muscle mass is dedicated to shoulder flexion vs extension, abduction vs adduction etc in the two species? i.e. if a flexor is absent in one species, is this reflected in lower flexor muscle mass overall, or is it simply that one of the other flexors is bigger and compensates and thus overall flexor mass remains similar? I did this kind of comparison in my analysis of birds vs crocs/lizards (Bates & Schachner 2012, Fig. 6A&B). Just a thought.

Minor comments
Abstract: “…we find striking similarities in muscle organisation and architectural parameters.” I feel “striking” is just a little sensational.

Abstract: “…distal muscle attachments are notably similar, while differences in proximal muscle attachments are driven by modifications to skeletal anatomy.” Are you talking about qualitative or quantitative differences in the attachment sites/areas? I would assume quantitative given you’re implying the level similarity is surprisingly high (I wouldn’t think qualitatively the muscle paths/sites would be expected to be particularly different would they?). Also, is “distal” the best word here? I would tend to think of wrist/manus muscles when distal is used, but you’re talking about upper arm muscles throughout.

Abstract: “….are statistically indistinguishable for an unexpected number of muscles.” I think this should be reworded. How many would one "expect", objectively, to be distinguishable statistically? You can’t say objectively as one person’s expectation may differ from another.

Introduction, line 52: I personally don’t like the language (“beyond recognition”) here as it seems a bit over the top, but it’s purely a personal preference.

Introduction, line 89: Charles et al. (2016) seems a slightly strange reference here given the sentence is concerned with modelling fossil taxa and estimating the necessary input parameters and this paper is about modelling extant mice gaits. There are quite a few simulation studies of extinct dinosaurs and hominids that could be referenced in this context.

Methods, general comment: Do you know anything about the specific animals used in the study? e.g. are they all wild, all captive, or a mixture? All adult? All male, female or mixed? Apologies if such things are stated and I’ve missed it.

Methods, Equation 1: When collating data on muscle architecture for my 2018 study on T. rex bite force (Bates & Falkingham 2018) I noted that some studies used this equation strictly regardless of whether pennation angle was 0 or >0. However, other studies used muscle-belly length (rather than fibre length) to calculate PSCA when the muscles were considered parallel-fibred (pennation = 0). Which approach are you using and why? We need to know this given you’re statistically comparing PSCA. Calculating it one way versus the other will impact on the data, and may impact on the results/conclusions.

Results, Lines 210-218: I think the description of limb bone orientation would be aided by a ‘purpose-built’ comparative figure of the 3D whole limbs, either in the main text or supplementary information.

Discussion, Line 509: When you say “overwhelmingly similar” do you mean purely qualitatively similar? Basically, origins/insertions/paths? I’m not personally “overwhelmed” by this myself (terrestrial mammals don’t differ that much do they?), but okay. I wouldn’t, however, be okay, with “overwhelmingly” being used if you’re also including the quantitative data in this statement. If it is just the qualitative morphology you're referring to then could you add “topologically” or “qualitatively” to this sentence.

Discussion, Lines 521-534: The paragraph starts to deal with the big issue underpinning the paper: how adapted, functionally, are the muscles in the extant taxa and what does this mean for reconstructing them in fossil taxa. However, the message that comes through here is not quite as clear or explicit as it could be I feel. The paragraphs starts as if the message is going to be “actually the muscles are quantitatively very similar – the similarities outnumber the differences.” You rightly go on to qualify this by explaining that the situation is more complicated than similar PCSAs equals similar function. But then there’s no explicit conclusion. Nor does one really follow in the next paragraph. What’s the explicit answer to the question you yourself pose – does functional similarity parallel anatomical similarity or not? I think there are quite important functional differences here (which you later point out in a different sub-section, lines 599-601) but the overall feeling one is left with at the end of the paper is that, overall, anatomy and function are “overwhelmingly” similar, and by inference we’re in a relatively straightforward position when it comes to taking the next step: reconstructing the muscles in fossil mammals.

Discussion, Line 619-622: “As non-mammalian synapsids are typically reconstructed as terrestrial generalists, the architectural parameters of the tegu and opossum may provide realistic “bookends” for the phylogenetically- and morphologically-intermediate non-mammalian synapsids.” What is a “terrestrial generalist” in this context? Presumably the ultimate over-arching goal here is to reconstruct how sprawling vs upright the transitional fossils were and constrain their maximal locomotor performance. I’m struggling to reconcile what is meant by “terrestrial generalist” in the context of having modern analogues to deliver these goals.

Discussion, Line 622: I’m being pedantic now but I don’t think you can really justify saying that what you’ve done with Massetognathus is a “test case” because you haven't strictly “tested” if the muscle parameters you’ve derived are capable of yielding viable/stable locomotion, or even that the muscle fascicles are of a length to allow them to contract across what might be considered a realistic length range for skeletal muscle. It’s semantics, but I’d rephrase to remove the implication that you’re actually testing the biomechanical plausibility of the reconstruction.

Discussion, Lines 644-658: This is a great paragraph. However, I think you could go a little further and discuss fibre lengths. And tendon lengths. One obvious issue with applying mass-scaled fibre (and tendon) lengths to an animal with a different size and/or shaped skeleton (=different muscle-tendon unit lengths) is tuning. You may, potentially, end up in a situation where the fibres are implausibly short or long given the force-length properties of skeletal muscle, and a muscle (or muscles) are essentially useless. I’ve had this issue within extant species where I’ve used muscle values from the literature and applied them to a musculoskeletal model built using a skeleton from another individual. While some of this is probably a mixture of allometry, ontogenetic differences and measurement error, I think it’s confounded further by the way we tend to model muscles as a single Hill-type fibres which leads to a bit of an all or nothing situation. I’m not suggesting you get into muscle modelling, but some explicit mention of the tuning issue would be useful for the palaeontological readership, and to reinforce the reality that just scaling bracketing extant taxa is not guaranteed to work even if aspects of their muscle morphology are statistically similar.

·

Basic reporting

No comment.

Experimental design

No comment.

Validity of the findings

No comment.

Additional comments

Overall the manuscript is well written and seems and I have only minor comments for the authors.

Minor comments
Style: I suspect your referencing software is responsible for it, but where you name the author in a sentence before citing them, you probably don’t need the name again in the brackets e.g. Romer (Romer, 1944). Can just do Romer (1944), but please double check with the journal style.

Methods: I know isometry is assumed for the range of body masses that you dissected with all being fairly close in size, however all of the specimens in Lugol’s Iodine are smaller (presumably a constraint of the diffusion/time), and one of the tegus was far smaller (~1/2 mass of the dissected animals). Do you see much variation from the assumed isometry between muscle volumes, and muscle attachment areas in these individuals (allowing for individual variation, and slight shrinkage through the staining procedure)? These stained specimens are interesting individuals as most are close to the 1kg size which you are standardising measures down to for your dissected individuals.

Lines 158-178: For personal/professional interest, with your stains were you able to get muscles stained all the way to the bone? We have had issues mapping attachments through staining particularly for more tendinous insertions. By using the fleshy insertions you presumably avoided this issue?

Ln 186: explain PBS

Ln 383: Capitalise figs.

Ln 454-459: This feels like a figure legend rather than an in text paragraph. Perhaps it should be interwoven with the next paragraph.

Ln 528-531: This sentence is awkwardly phrased, possibly due to the “while pennation”. Consider splitting the sentence, e.g. “Muscle PCSA relates to the force requirements for body support and locomotion (Eng et al., 2008). Pennation is linked to PCSA and thus the force capacity for a given volume of muscle, but may also enable dynamic gearing of mechanical output (Azizi, Brainerd & Roberts, 2008).”

It may also be worth explicitly stating here the implications of your statements about the differences between the species. I.e. to get the same PCSA and similar pennation from different masses, and fascicle lengths you have either:

1) Parallel muscles that are short vs long between the species or,
2) Pennate muscles that are similar lengths but different volumes/masses or
3) Multipennate muscles vs pennate of different volumes/masses

This is what your numbers in Table 1 and descriptions say quite well. Your figures may also support this Didelphis bones being almost twice the size of Salvator so muscles may be longer just to cross the larger distance BUT you should mention in your figures which individual is figured as it might be unfair to be naively comparing the 0.6kg Salvator to a 1.11kg Didelphis.

Ln 540: Italicise Varanus exanthemicus.
Ln 549: Italicise S. merianae and D. virginiana.

Line 613: Is it possible that sprawling amniotes have a greater mobility of the humeral head relative to the glenoid fossa so require less stabilisation?

Line 640: How did you measure Massetognathus origin areas?

Figures: All are very hard to read even zoomed in on a computer screen when they embedded with text (e.g. a pdf) due to the size of the text. Also check your colours with some of the yellow text being potentially difficult to read. I would suggest checking on print-outs.

Tables: Whilst I appreciate you measured the pennation angle using ImageJ, I would caution against the level of precision you are claiming with tables showing 2-3 decimal places, particularly with any variation in camera angle, muscle “squishing” etc. going to have an effect on pennation angles in muscles that are no longer in situ. Consider reducing to 1 d.p.

Sup. Info: No legends? Also check formatting for table S1. Is it possible to make the table landscape?

Sup text: Check spacing between “which were” in m. scapulohumeralis line 3.

Raw data: Check what Excel has done to your specimen labels as Sep 71, 74, 85, 87, 88, 92 have all become dates. Either change formatting from general, or put an apostrophe before the text e.g. ‘Sep71 which will freeze it and the apostrophe disappears from standard view.

Also why are skin measurements included in the excel sheet? Please check what is in the spreadsheet, and whilst some of it is great data to have, it should probably have a caveat in the sup text or at the top of the excel file to explain its presence (and that of the muscles that may not have been discussed, and random photo notes too)

·

Basic reporting

The present research presents a functional myologic analysis of the musculature of two similar-sized species which constitute a phylogenetic and morphofunctional bracket for inferring four non-conserved myologic characters of a fossil species, Massetognathus pascuali. The manuscript is self-contained and presents relevant results as it is the first estimation of PCSA in a fossil species. This character is fundamental for a confident functional comparison among species and the authors provide a strong base for myologic parameter´s calculation by dissecting two similar-sized species as “brackets”. Although the general interest is not new (evolution of pectoral girdles in vertebrates, associated to habitat and locomotor changes has been largely studied), the approach is original and worthy of publication.

The paper has a good structure that allows easy reading and interpretation, also, structure is ok according the Journal requested format. The English writing is good and unambiguous. Although I am not a native English speaker, I found three or four things that did not sound correct, so I left some suggestions in the attached file. Please, make sure a native speaker checks on them.

Cited literature was pertinent and sufficient. Regarding background, I believe the authors could add some small details to help scientist who are not specialist in vertebrate anatomical evolution, better understand the anatomical differences between species (particularly pectoral girdle). Also, no functional description is made of the fossil about its posture and locomotor mode, more information about this is needed.

Article´s structure is ok and most raw data is available in well-structured tables, but I was not able to find availability of scanned 3D images throughout the manuscript.

My only suggestion about Table 1 is to mark significant differences (may be the usually used *), even when they are in Fig. 6, as this information helps readers.

Figures are all necessary, in a general point of view they are appropriately described and labeled, but I would suggest the following changes:

All figures: The journal format ask for labeling each part of a multi part figure with an uppercase letter, this requisite is not present in the figures. Also, as requested by the journal, Figures must be cited in the order they appear in the text, please check that the last figures appear cited first in your manuscript (see also comments in attached file).

Please see other specific changes in the attached file

Additionally, I would suggest if possible, another figure (should be the first or in supplementary material) with osteological structure compared between the two studied species and the fossil, again to make it easier to read and understand for a non-specialized vertebrate anatomist, but also to make a more complete functional reading of your findings. Alternatively, authors could consider adding in Fig. 1, the names of the bones mentioned in the text (may be with coding) and address the reader to Fig. 1 in Lai et al., 2018 (doi: 10.1111/joa.12766) to see the osteological structure of the fossil species.

All results are the ones relevant to the hypothesis.

Experimental design

Primary research falls within Aims and Scope of the Journal. The manuscript presents a clear question, i.e. calculating non-conserved myologic parameters like PCSA in fossil species. Methods were described with sufficient detail & information to replicate.

Validity of the findings

The data on which the conclusions are based are statistically sound and controlled. Conclusions are well stated, linked to original research question and extremely limited to supporting results. Actually, I would like the authors to use a bit more of the Journal´s allowance for speculation to see more interpretations with previous analysis. For example, how do you think your knew approach influences previous morphofunctional interpretations of the fossil species? (I mean besides the posture, are there other ecological factors that can be associated to these changes?) And in a broader way? How would the possibility of calculating these parameters will affect future fossil´s research?

Finally, results are strong and with biological sense (not only statistical), and as a first approach to solving an extremely difficult problem, it is well sustained and practical.

Additional comments

This is a well-supported study that provides a new approach to solving an important issue in inferring fossil species eco-morphology. It provides a practical and useful example on how to calculate important myologic parameters fundamental to complement typical fossil´s reconstructions.
A have mostly small suggestions to make about the paper and they are disclosed in the attached file, but here I summarize the most notable ones.

1) The paper is very well written, but in a certain way I feel the authors are writing for a very specialized public, I would suggest some small changes that can make reading easier for non-specialists, e.g., adding the names of bones in figures (or a new figure; see attached file) or adding more background of Massetognathus pascuali inferred ecology and ecomorphology.
2) Discussion section is extremely limited to supporting results. I would like the authors to use a bit more of the Journal´s allowance for speculation to see more interpretations including information of previous analysis.
3) I don´t know if I missed it, but I haven’t seen availability of the 3D scanned images (as requested by the Journal)
4) Improving size letters in figures and change format according to the Journal

---

## Round 0.2 · Minor Revisions

Thank you for your consideration of our suggestions. I need to see, however, a full justification or, even better, an examination of muscle lengths to calculate PSCA. As our first reviewer has highlighted, this is not only a semantic issue, so please take it in full consideration. Our second reviewer pointed out a few minor changes that need your attention.
If you do not mind, please correct the reference in the text as I put it here:
Montero R, Abdala V, Moro SA, Gallardo G. 2004. Atlas de Tupinambis rufescens (Squamata: Teiidae). Cuadernos de Herpetología 18 (1): 17-32.

·

Basic reporting

Absolutely fine.

Experimental design

Absolute fine.

Validity of the findings

Fine.

Additional comments

Many thanks to the authors for responding so comprehensively and addressing the queries I raised about the first submission. As I noted in my first review, a lot of my points were suggestions rather than issues or obligatory ‘corrections’ and so I’m fine with the authors arguments not to implement some of them. The new Fig 1 is an excellent addition!

I think the only thing left to resolve with me is the issue around muscle length vs fibre length to calculate PCSA in parallel fibred muscles (Equation 1). I fully appreciate that this is going to feel like a very annoying niggle to the authors, but this issue also needs to be confronted in a very similar paper by the authors that I am reviewing for a different journal. So we might as well deal with it here and hopefully kill two birds with one stone.

Firstly, please could the authors remove the citation of Bates and Falkingham 2018 from the following sentence:
“We therefore opted to follow the common practice of applying Eq.1 to all muscles, regardless of pennation (Bates & Falkingham, 2018).”

I feel citing us here implies that we are advocating that fibre length should be used to calculate PCSA when pennation = 0. That may not be what the author were trying to do by citing us, but I feel that it carries that implication and I’d rather avoid it (because we don’t advocate it. We sit on the fence in that paper). Secondly, I’m not completely satisfied with the justification added to support the authors choice of using fibre length. This is partly because I think that, in principle, the justification should be based around consideration of the links between muscle architecture and force generating capacity, rather than simply what was or wasn’t convenient to measure. I accept that in science we sometimes have to pick less accurate but more practical options, but in those circumstances, we should explain why that sub-optimal method is the best we can do and what effect it might have on the data/conclusions. So, are the authors saying that based on the links between muscle architecture and force generating capacity we should be using muscle belly length, but they can’t do that here because measuring muscle length is too subjective? That might be okay (i.e. I would just ask the authors to provide an expanded and more explicit explanation of the situation), except in the other paper I’m reviewing by the authors (same methods, same muscle groups, but additional species) they have included an analysis of muscle fibre length:muscle length in forelimb muscles. The fact they are measuring and analysing muscle length values for the same muscles in another paper makes it difficult for me to accept the justification they’ve added here (“we can’t meaningfully measure muscle length”) for using fibre length instead of muscle length to calculate PCSA. As the authors will see from my comments on the other paper, I lean towards using muscle length rather than fibre length because the forces do not sum in parallel fibred muscles in the same way that they do in pennate muscles. But I’m not a muscle physiologist. If the authors can counter this with a stronger theoretical argument about how muscle architecture = force then that’s fine. Please do so and include that in the paper. I’m not trying to force the authors to use muscle length, but I do think in light of the above they need better justification for using fibre length than they’ve offered in their resubmission.

·

Basic reporting

The authors have made most of the changes I requested and I think the manuscript is pretty much ready for publication. I have a couple of very minor formatting things that should be corrected.

ln 660: spelling - gsupraspinatus
Figure 7: Numbers from the tests for significance are still very small and difficult to read.
Raw Data: Some of your specimen numbers are still in date format. See previous review comments for how to fix.

Experimental design

No comment

Validity of the findings

No comment

---

## Round 0.3 · accepted · Accept

Our first reviewer is happy with your last explanations about the issue of fibre length vs muscle length. It seems to me that it remains an open and interesting question. The ideas interchanges that happened in relation to this point are quite useful for other readers and I hope that the authors allow them to have access to it.

Please note that the quote about the Atlas of Tupinambis in the text should be corrected on page 4 ln. 130: Montero, Abdala & Moro, 2004

·

Basic reporting

Fine

Experimental design

Fine

Validity of the findings

Fine

Additional comments

I’d like to thank the reviewers for indulging me in the issue of muscle length vs fibre length when calculating PCSA. I asked for a justification and they’ve provided one so I’m happy to see the paper published in its current form.

To explain my mild obsession with this issue: Two years ago I was asked by two reviewers to justify using fibre length rather than muscle length to calculate PCSA when pennation = 0. I responded in near identical fashion to the way you have here. My paper was rejected because both reviewers subsequently responded with a mechanistic argument to support the idea that it should be muscle length rather than fibre length. I’ve since spoken to several biomechanicists and muscle physiologists that agree. As I’ve said, I’m leaning this way now too, but its something I want to look further into myself with data. The rationale is as follows. Equation 1 (calculation of PCSA) is usually followed by the equation

Muscle force = PCSA x Force Per Unit Area (Equation 2).

Equation 2 is why we bother to calculate PCSA, even if (as the current study) we don’t actually carry out the calculation directly. If pennation = 0 then either:

(1) fibres run in parallel from origin to insertion (=muscle length; e.g. muscle length = 10cm, fibre length = 10cm),
(2) they are arranged ‘perfectly’ in parallel end-to-end and when added up they equal muscle length (e.g. muscle length = 10cm, 2 sets of fibres 5cm long arranged end-to-end to make up the 10cm), or
(3) the fibres are partly arranged in series and partly overlap.

If muscle fibres work in series (fully or partly), the muscle model presented in equations 1 and 2 becomes an abstraction as it assumes muscle fibres do not to work in series. For fibres arranged in series it is therefore probably more realistic, given equations 1 & 2, to treat them as spanning muscle length. Equation 1 (and then 2) allow us to take into account the greater number of shorter muscle fascicles that can be packed into a pennate muscle. These fascicles act in parallel so their forces sum. An assumption of the above PCSA and force calculation is that the individual fascicles run the full length of non-pennate muscle and from aponeurosis to aponeurosis in a pennate muscle. If there are fibers or even fascicles arranged in series (as they must be in some your muscles), they do not increase PSCA or muscle force in the linear way that results from using fibre length (instead of muscle length) in equation 1. I found these two papers quite useful when thinking about the issue after my paper was rejected.

Anapol F, Shahnoor N, Ross CF. 2008. Scaling of reduced physiologic cross-sectional area in primate muscles of mastication. In: Vinyard C, Wall CE, Ravosa MJ, editors. Primate Cranio-facial Function and Biology. New York: Springer. pp 201–216.

Infantolino, B.W., Neuberger, T. and Challis, J.H., 2012. The arrangement of fascicles in whole muscle. The Anatomical Record, 295(7), pp.1174-1180.

Anyway, congratulations on a great study.